# Zeus: Towards Tuning-Free Foundation Model for Time Series Analysis

Yisong Fu [1 2]  Zezhi Shao [1]  Chengqing Yu [1]  Yujie Li [1 2]  Yongjun Xu [1 2 3]  Xueqi Cheng [1 2]  Fei Wang [1 2]

## Abstract

We present ZEUS, a unified tuning-free Time Series Foundation Model (TSFM) that delivers superior performance across diverse analysis tasks without any task-specific fine-tuning. Unlike prior studies that primarily focus on zero-shot forecasting but require task-specific tuning for other tasks, ZEUS bridges this gap by addressing two fundamental challenges in multi-task generalization. First, to reconcile point-level granularity with long-sequence scalability, ZEUS incorporates a multi-scale Transformer featuring point-wise tokenization and a U-shaped hierarchy, effectively balancing fine-grained fidelity with computational efficiency. Second, to accommodate varying inductive biases across different tasks, ZEUS introduces Multi-Objective Temporal Masking (MOTM), a unified strategy that supports heterogeneous tasks (*e.g.*, extrapolation, interpolation, and global abstraction) within a single framework. Extensive experiments across five representative tasks demonstrate that ZEUS consistently achieves competitive results in tuning-free settings, underscoring its potential as a general-purpose TSFM. The code is available at https://github.com/GestaltCogTeam/Zeus.

## 1. Introduction

Time series analysis is a pivotal field with broad-ranging applications, from weather forecasting (Fu et al., 2025) and physiological anomaly detection (Šabić et al., 2021), to data imputation (Luo et al., 2018) and human activity recognition (Vrigkas et al., 2015). Its profound practical utility continues to attract significant research and industrial interest.

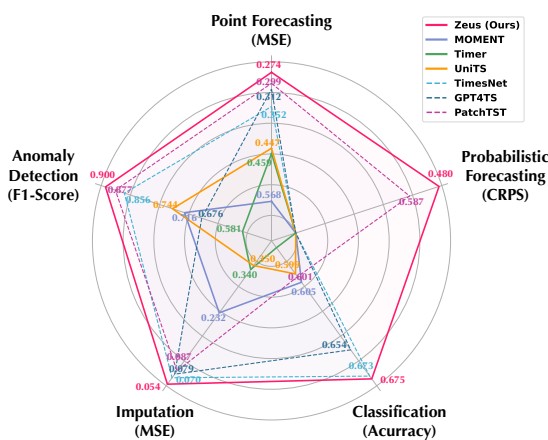

Figure 1. Overall performance comparison of ZEUS under the tuning-free setting. ZEUS surpasses full-shot task-specific models (*dashed lines*) and significantly outperforms other TSFMs in tuning-free setting (*solid lines*).

Inspired by the success of foundation models in language (OpenAI, 2023), images (Ramesh et al., 2021), and videos (Liu et al., 2024c), researchers have been striving to develop general-purpose time series foundation models (TSFMs). Trailblazing efforts like MOMENT (Goswami et al., 2024), Timer (Liu et al., 2024b), and UniTS (Gao et al., 2024) have shown the potential of unified architectures pretrained on large-scale datasets to address multiple downstream tasks.

Despite these advances, such models have primarily demonstrated zero-shot capability only in forecasting tasks, while adapting to other downstream tasks typically requires additional training (Liu et al., 2024b; Gao et al., 2024). This limitation contrasts with the general expectations of foundation models and has consequently led most recent efforts to focus predominantly on forecasting (Shi et al., 2025; Cohen et al., 2025; Auer et al., 2025). This limitation can be traced to two fundamental dilemmas in existing TSFMs.

❶ **Architectural dilemma between granularity and scalability.** Most existing TSFMs adopt patch-wise tokenization (Nie et al., 2023) to improve semantic density and computational efficiency. However, this design inevitably sacrifices point-level temporal details, hindering its effectiveness on reconstruction-based tasks like imputation and anomaly detection. For example, MOMENT, pretrained with patch-level reconstruction, exhibits a significant performance drop

[1]State Key Laboratory of AI Safety, Institute of Computing Technology, Chinese Academy of Sciences [2]University of Chinese Academy of Sciences [3]Xiamen Institute of Data Intelligence. Yisong Fu <fuyisong24s@ict.ac.cn>. Correspondence to: Fei Wang <wangfei@ict.ac.cn>.

*Proceedings of the 43rd International Conference on Machine Learning*, Seoul, South Korea. PMLR 306, 2026. Copyright 2026 by the author(s).

when shifting from patch-missing to point-missing imputation (evidenced in Table 6). In contrast, point-wise tokenization (Ansari et al., 2024; Shi et al., 2025) preserves fine-grained structure but suffers from low information density and prohibitive computational overhead on long sequences.

❷ **Training dilemma for divergent inductive biases.** Although recent works attempt a unified modeling paradigm, they often neglect the fundamentally different inductive biases required by each objective: forecasting relies on extrapolation, imputation and anomaly detection necessitate interpolation, and classification requires global abstraction. This heterogeneity indicates that a single, monolithic training objective, such as BERT-style masked reconstruction (Goswami et al., 2024; Zhang et al., 2025) or GPT-style autoregressive generation (Liu et al., 2024b), is unlikely to simultaneously endow all necessary capabilities.

Collectively, these dilemmas have hindered prior TSFMs from fully generalizing across diverse tasks, often necessitating task-specific fine-tuning to achieve strong performance. To address these challenges, we introduce ZEUS, a unified TSFM that bridges point-level fidelity and long-sequence scalability while supporting diverse task-specific inductive biases within a single pretraining framework. ZEUS is designed to operate in a fully tuning-free [1] manner, enabling out-of-box deployment across diverse downstream tasks.

To overcome the tension between granularity and scalability, ZEUS adopts a U-shaped multi-scale hierarchy with point-wise tokenization. It follows a *fine-to-coarse-to-fine* information flow, progressively aggregating fine-grained information into coarse-grained latent representations and then symmetrically refining them. Lightweight Transformer blocks are used at high resolutions to preserve local details, while deeper and wider blocks operate on compressed representations to efficiently capture global dependencies.

Complementing this architecture, we address the challenge of divergent inductive biases with MOTM, a Multi-Objective Temporal Masking strategy that jointly trains ZEUS for extrapolation, interpolation, and global abstraction. By exposing the model to predictive, point-wise, and block-wise corruption patterns during pretraining, MOTM enables ZEUS to learn a unified yet versatile representation space that supports out-of-the-box performance across diverse downstream tasks.

Experimentally, ZEUS achieves the consistent state-of-the-art performance in a tuning-free manner across widely recognized benchmarks for five downstream tasks (Figure 1), including long time series benchmark (Wu et al., 2023) for point forecasting and imputation, GIFT-Eval (Aksu et al., 2024) for probabilistic forecasting, UCR anomaly archive

(Wu & Keogh, 2021) for anomaly detection, and UEA archive for classification. These results demonstrate the strong generalizability of ZEUS across diverse tasks. In summary, our contribution is three-fold:

- We present ZEUS, a unified multi-scale Transformer that preserves point-level fidelity while efficiently modeling high-level semantics over long sequences.

- We introduce MOTM, a multi-objective temporal masking strategy that aligns diverse task-specific inductive biases within a single pretraining framework.

- To the best of our knowledge, ZEUS is the first TSFM to achieve competitive performance across five downstream tasks without task-specific adaptation, as validated on well-established benchmarks.

## 2. Related Work

### 2.1. Task-Specific Time Series Model

Time series analysis plays a fundamental role in a wide range of applications, including forecasting, imputation, anomaly detection, and classification. Early statistical and traditional machine learning methods typically rely on case-by-case fitting, facing limitations when applied to large-scale data. With the advancement of deep learning, task-specific neural models have become the dominant paradigm (Xu et al., 2021; Huang et al., 2025). CNN-based models, such as TS2Vec (Yue et al., 2022), TimesNet (Wu et al., 2023) and ModernTCN (donghao & xue, 2024), have shown strong potential to support multiple tasks within a unified framework. Transformer variants (Wu et al., 2021; Eldele et al., 2024; Nie et al., 2023; Liu et al., 2024a) further enhance the ability to capture long-range dependencies and complex temporal patterns. Inspired by advances in LLMs, recent work has explored transferring pretrained language models to time series tasks (Gruver et al., 2023; Zhou et al., 2023; Jin et al., 2024). Typically, GPT4TS (Zhou et al., 2023) explores the use of pretrained GPT-2 as a frozen backbone for time series analysis, adapting it to downstream tasks through lightweight output projections. However, these methods still rely on task-specific training for each downstream application.

### 2.2. Time Series Foundation Models

Building upon task-specific precursors, recent research has increasingly explored pretraining foundation models on large-scale time series corpora to enhance cross-task generalization, with Transformer (Vaswani et al., 2017) becoming the predominant architecture. Representative approaches such as MOMENT (Goswami et al., 2024) and TimesBERT (Zhang et al., 2025) adopt BERT-style masked reconstruction objectives to learn contextualized representations by

---

[1] In this paper, the term *tuning-free* refers to performing inference without tuning or retraining any model parameters.

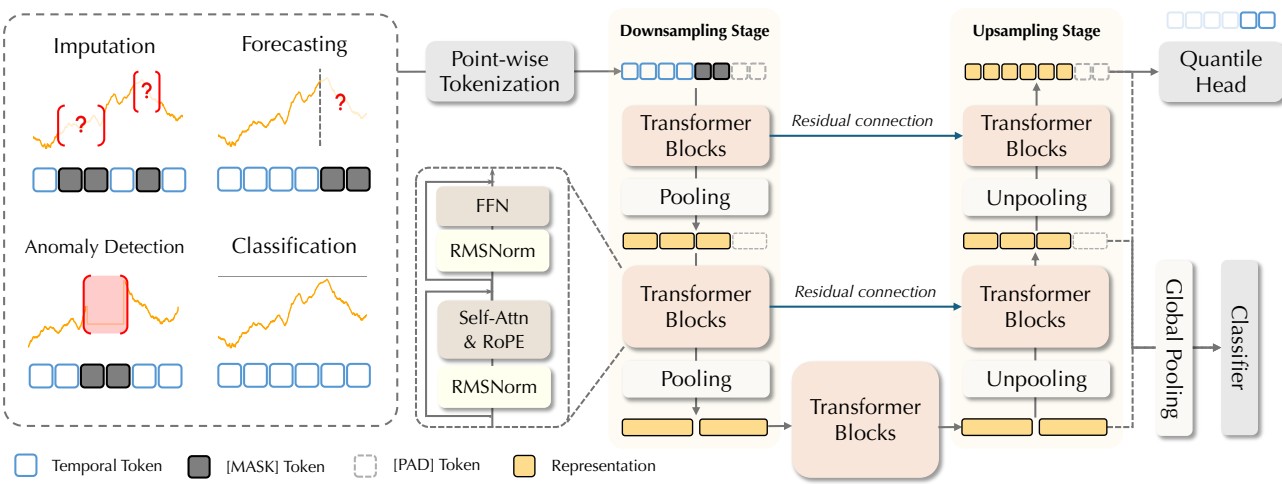

*Figure 2.* Overall architecture of ZEUS. Inputs from different downstream tasks are first unified into a common format and converted into point-wise tokens via tokenization. The resulting sequence is then processed by a U-shaped multi-scale Transformer. Quantile head is used to produce probabilistic outputs, while for classification tasks, global pooling is applied to obtain sequence-level representations.

recovering masked patches. UniTS (Gao et al., 2024) extends this paradigm by incorporating task prompts, facilitating flexible task adaptation. In contrast, Timer (Liu et al., 2024b) adopts a GPT-style autoregressive formulation to cast diverse time series tasks into a unified generative framework. Despite their progress, existing TSFMs remain constrained by architectural and training-design limitations, which hinder fine-grained modeling and necessitate task-specific adaptation in practice.

## 3. Zeus

### 3.1. Overall Framework

ZEUS is a multi-scale encoder-only Transformer architecture for general-purpose time series analysis. As illustrated in Figure 2, it adopts point-wise tokenization to preserve fine-grained temporal resolution, and processes the resulting tokens through a symmetric downsampling–upsampling hierarchy of Transformer encoders. Representations are progressively aggregated from fine to coarse scales and then refined back to fine resolution. This design resolves the dilemma between granularity and scalability: it preserves point-level details required by reconstruction-oriented tasks while promoting semantically compact and computationally efficient representations for large-scale pretraining.

### 3.2. Tokenization

ZEUS adopts a point-wise tokenization scheme, where each time step is treated as an individual token. We conduct pretraining in a univariate paradigm and employ the channel-independent strategy to handle multivariate time series (Nie et al., 2023). Given a time series $\mathbf{x} = \{x_1, x_2, \cdots, x_T\}$, we first apply instance normalization (Kim et al., 2021) to

remove scale variations. Then each time step is mapped to a hidden representation via a gated embedding layer:

$$\mathbf{h}_t^{(0)} = \mathbf{W}_r x_t + \mathbf{W}_d\big(\sigma(\mathbf{W}_g x_t) \odot \mathbf{W}_u x_t\big), \qquad (1)$$

where $\sigma(\cdot)$ denotes a nonlinear activation. This design increases the expressive capacity of token embeddings.

We further introduce two learnable tokens: a [MASK] token for masked reconstruction and a [PAD] token for variable-length sequences and multi-scale alignment, enabling unified handling of masking, missing values, and padding. As shown in Figure 2, by replacing temporal tokens with the [MASK] token, diverse downstream tasks can be uniformly formulated under a masked modeling paradigm, allowing ZEUS to support multiple tasks within a unified framework. In forecasting tasks, several [MASK] tokens are appended after the historical window; in imputation tasks, missing values are replaced with [MASK]; in anomaly detection tasks, the target segment can be replaced with [MASK], and reconstruction or prediction errors are used as anomaly scores.

### 3.3. Multi-Scale Architecture

While point-wise tokenization preserves temporal fidelity, it introduces substantial computational challenges for long sequences. ZEUS resolves this tension through a U-shaped hierarchy that progressively adjusts temporal resolution across symmetric scales $\{s_1, s_2, \cdots, s_K\}$, where $s_i = s_{K-i+1}$ defines the number of time steps aggregated into a token.

**Downsampling and Upsampling Stages** This U-shaped architecture consists of a downsampling stage to compress fine-grained information into high-level semantics via pooling operation, and a symmetric upsampling stage, which

employs unpooling to progressively restore local details.

In the downsampling stage, given the hidden states $\mathbf{h}^{(i)} \in \mathbb{R}^{T_i \times d_i}$ at scale $i < \frac{K-1}{2}$, a pooling layer aggregates $r = s_{i+1}/s_i$ adjacent tokens into a single representation. The pooled representations are then processed by a Transformer encoder to model higher-level temporal dependencies.

$$\mathbf{p}^{(i)} = \text{Reshape}(\mathbf{h}^{(i)}, \mathbb{R}^{\frac{L_i}{r} \times (rd_i)})\mathbf{W}_p, \qquad (2)$$

$$\mathbf{h}^{(i+1)} = \text{TrfmEncoder}(\mathbf{p}^{(i)}), \qquad (3)$$

where $\mathbf{W}_p \in \mathbb{R}^{(rd_i) \times d_{i+1}}$ is a learnable linear projection.

Conversely, in the upsampling stages ($i \geq \frac{K-1}{2}$), an unpooling layer expands the sequence and integrates high-resolution features from the corresponding scale via residual skip connections:

$$\mathbf{P}^{(i)} = \text{Reshape}(\mathbf{h}^{(i)}\mathbf{W}_u, \mathbb{R}^{(rL_i) \times d_{i+1}}) + \mathbf{h}^{(K-i+1)}. \quad (4)$$

The resulting representations are then fed into a Transformer encoder to refine fine-grained temporal details. In this architecture, lightweight Transformer blocks operate at fine scales to capture local patterns, while deeper and wider blocks process coarser representations to model long-range dependencies. This multiscale design enables ZEUS to efficiently balance temporal fidelity and scalability.

**Transformer Block**   Each Transformer block employs multi-head self-attention with rotary positional embeddings (Su et al., 2024), together with gated feed-forward networks (Shazeer, 2020). To enhance training stability, we adopt RMSNorm (Zhang & Sennrich, 2019) and a pre-LN scheme (Xiong et al., 2020). All attention modules are implemented with FlashAttention v2 (Dao, 2024), enabling memory-efficient and scalable training on long sequences.

**Quantile Head**   ZEUS employs a quantile head that provides $|Q|$ quantile values for each time step, enabling probabilistic reconstruction instead of point estimation. Formally, the head implements a mapping $\mathbb{R}^{d_K} \rightarrow \mathbb{R}^{|Q|}$. In our implementation, ZEUS predicts a set of nine quantile levels $Q = \{0.1, 0.2, ..., 0.9\}$. For classification task, we directly use the globally pooled representations.

### 3.4. Multi-Objective Temporal Masking

The key challenge for TSFMs to generalize across tasks lies in the distinct inductive biases required by different objectives. Forecasting requires extrapolation, whereas imputation relies on interpolation. Anomaly detection and classification introduce more complex demands: point anomalies require sensitivity to local variations, while contextual

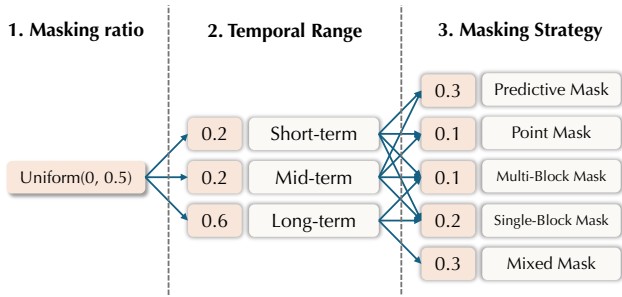

*Figure 3.* The MOTM pipeline. MOTM hierarchically determines the masking ratio, scales the temporal scope, and applies diverse masking strategies to jointly optimize for extrapolation, interpolation, and local-global feature extraction.

anomalies necessitate modeling global consistency (Liu et al., 2025a). Moreover, classification calls for both global abstraction and the identification of local shapelets (Le et al., 2022). However, current TSFMs typically employs a simple BERT-style (Goswami et al., 2024; Zhang et al., 2025) or GPT-style objective (Liu et al., 2024b), which fails to simultaneously capture these heterogeneous inductive biases.

To bridge this gap, we introduce multi-objective temporal masking (MOTM), a unified masking-based training strategy that equips ZEUS with heterogeneous inductive biases. Figure 3 illustrates the pipeline of MOTM. For each instance, the overall corruption ratio is sampled from $\mathcal{U}(0, 0.5)$ to enhance robustness to varying levels of missing. We then select a temporal scope that jointly cover short-, mid-, and long-term dependencies. Finally, a masking scheme is sampled from a diverse set, including predictive, point, multi-block, and single-block masking, as well as their combinations. Below, we detail each masking strategy and discuss its role in shaping task-relevant inductive biases.

**Predictive Mask**   To bolster extrapolative capacity essential for forecasting, we adopt a predictive masking scheme that masks a suffix of the sequence. Specifically, given a probability $p$, we mask the last $L_p = \lfloor Tp \rfloor$ time steps, requiring the model to learn temporal causality and capture long-range dependencies beyond the observed horizon.

**Point Mask**   This widely-used strategy targets point-wise interpolation by randomly masking individual time steps. This fine-grained corruption forces the model to leverage local continuity and short-range correlations, while serving as a regularizer against overfitting to specific patterns.

**Multi-Block Mask**   To simulate the contiguous missing-ness prevalent in real-world deployments, we introduce a multi-block masking strategy. Inspired by span-based corruption used in language modeling (Joshi et al., 2020; Lewis et al., 2020), this method shifts the objective from point-

*Table 1.* Zero-shot point forecasting results, averaged over four prediction lengths $\{96, 192, 336, 720\}$. Datasets used during pretraining are excluded from evaluation for the corresponding models and are denoted by a dash ($-$). Best and second-best results are shown in **bold** and underlined, respectively. Full results are provided in Table 7.

| Models | Time Series Foundation Models (Zero-shot) | | | | | | | | Pretrained Forecasting Models (Zero-shot) | | | | | | | | | | Task-Specific Models (Supervised) | | | | | | | |
| | ZEUS (Ours) | | MOMENT (2024) | | Timer (2024b) | | UniTS (2024) | | Kairos$_l$ (2025) | | Toto$_b$ (2025) | | Sundial$_l$ (2025b) | | Time-MoE$_l$ (2025) | | ChronosBolt$_b$ (2024) | | ModernTCN (2024) | | GPT4TS (2023) | | TimesNet (2023) | | PatchTST (2023) | |
| Metric | MSE | MAE | MSE | MAE | MSE | MAE | MSE | MAE | MSE | MAE | MSE | MAE | MSE | MAE | MSE | MAE | MSE | MAE | MSE | MAE | MSE | MAE | MSE | MAE | MSE | MAE |
|---|---|---|---|---|---|---|---|---|---|---|---|---|---|---|---|---|---|---|---|---|---|---|---|---|---|---|
| ETTh1 | **0.377** | **0.399** | 0.715 | 0.580 | 0.499 | 0.463 | 0.496 | 0.478 | 0.427 | 0.410 | 0.435 | 0.413 | 0.395 | 0.420 | 0.394 | 0.420 | 0.479 | 0.429 | 0.419 | 0.432 | 0.438 | 0.437 | 0.485 | 0.469 | 0.427 | 0.437 |
| ETTh2 | **0.320** | 0.364 | 0.394 | 0.428 | 0.413 | 0.419 | 0.427 | 0.431 | 0.350 | 0.374 | 0.340 | **0.363** | 0.334 | 0.387 | 0.405 | 0.415 | 0.341 | 0.364 | 0.346 | 0.392 | 0.405 | 0.433 | 0.422 | 0.425 | 0.361 | 0.402 |
| ETTm1 | **0.322** | **0.359** | 0.714 | 0.554 | 0.837 | 0.593 | 0.690 | 0.538 | 0.348 | 0.365 | 0.378 | 0.396 | 0.331 | 0.369 | 0.376 | 0.405 | 0.395 | 0.368 | 0.346 | 0.376 | 0.357 | 0.383 | 0.458 | 0.428 | 0.346 | 0.376 |
| ETTm2 | **0.249** | 0.305 | 0.359 | 0.388 | 0.373 | 0.388 | 0.328 | 0.362 | 0.252 | **0.303** | 0.267 | **0.303** | 0.254 | 0.315 | 0.258 | 0.315 | 0.278 | 0.307 | 0.265 | 0.322 | 0.275 | 0.331 | 0.286 | 0.328 | 0.256 | 0.312 |
| ECL | **0.157** | **0.243** | 0.900 | 0.762 | 0.304 | 0.362 | 0.449 | 0.490 | - | - | 0.161 | **0.243** | 0.166 | 0.262 | - | - | - | - | 0.163 | 0.259 | 0.168 | 0.263 | 0.198 | 0.301 | 0.167 | 0.262 |
| Weather | **0.217** | 0.247 | 0.326 | 0.353 | 0.326 | 0.342 | 0.291 | 0.306 | 0.231 | 0.253 | 0.224 | **0.245** | 0.238 | 0.275 | 0.256 | 0.288 | 0.237 | 0.254 | 0.232 | 0.270 | 0.230 | 0.263 | 0.261 | 0.286 | 0.236 | 0.275 |
| # Wins | **19** | **14** | 0 | 0 | 0 | 0 | 0 | 0 | 3 | 1 | 0 | 6 | 0 | 0 | 0 | 0 | 1 | 5 | 1 | 0 | 0 | 0 | 0 | 0 | 0 | 0 |

wise reconstruction to structured recovery. However, instead of adopting the Poisson distribution commonly used in language modeling, we employ a uniform distribution, which poses a more challenging setting for time series. Specifically, given a masking budget $L_p = \lfloor Tp \rfloor$, we sample block lengths $\{\ell_k\}$ from $\ell_k \sim \mathcal{U}(1, 24)$, until $\sum_k \ell_k \approx L_p$, and randomly distribute the blocks along the sequence. This strategy compels the model to perform interpolation under structured missingness rather than simple local smoothing.

**Single-Block Mask**   Complementary to multi-block masking, we introduce a single-block masking strategy that removes one long contiguous segment from an arbitrary position in the sequence. This strategy encourages the model to maintain global consistency, which is crucial for classification and contextual anomaly detection.

**Mixed Mask**   We further employ a mixed masking scheme that combines multiple masking strategies to increase the difficulty of the pretraining objective. In practice, simpler masks (multi-block and point mask) are paired with harder ones (predictive and single-block mask), ensuring that mixed masking remains challenging yet learnable.

**Training Objective**   Under all masking strategies, ZEUS is trained in a masked reconstruction manner with the quantile loss. Given a binary mask $\mathcal{M} \in \{0, 1\}^T$, the loss is computed only on masked positions:

$$\mathcal{L} = \frac{1}{|Q||\mathcal{M}|} \sum_{t:\mathcal{M}_t=1} \sum_{q \in Q} \begin{cases} q(y_t - \hat{y}_t^q), & \hat{y}_t^q \le y_t, \\ (1-q)(\hat{y}_t^q - y_t), & \hat{y}_t^q > y_t. \end{cases} \quad (5)$$

Here, $\hat{y}_t^q$ denotes the prediction at quantile level $q$.

### 3.5. Pretraining Data

We train ZEUS on a large-scale corpus of real and synthetic time series, comprising about 300B observations collected from diverse domains and frequency. The real-world data are mainly sourced from the Chronos datasets (Ansari et al., 2024) and the GiftEvalPretrain datasets (Aksu et al., 2024). To enhance pattern diversity beyond real-world records, we additionally construct Aegis-Syn, a synthetic dataset that extends KernelSynth (Ansari et al., 2024) with richer temporal structures, particularly non-smooth and discontinuous patterns that are underrepresented by Gaussian process–based generators. To mitigate pattern imbalance in the pretraining corpus, we adopt a balanced sampling strategy (Shao et al., 2025a). All evaluation datasets are excluded from pretraining to prevent data leakage. Detailed descriptions of the pretraining data and Aegis-Syn are provided in Appendix E.

## 4. Experiments

Based on the above approach, we implemented and trained ZEUS with five scales and 12 Transformer layers, comprising approximately 100M parameters in total. Details are provided in Appendix A. In this section, we conduct a comprehensive evaluation on well-established benchmarks to assess ZEUS's performance across five downstream tasks: point forecasting (§4.1), probabilistic forecasting (§4.2), imputation (§4.3), anomaly detection (§4.4), and classification (§4.5). See Appendix D for detailed experimental settings and baseline descriptions.

### 4.1. Point Forecasting

**Setups**   We adopt six widely used long-term forecasting benchmarks (Wu et al., 2023; Shao et al., 2024) to evaluate ZEUS's zero-shot performance in point forecast-

*Table 2.* Performance evaluation on the GIFT-Eval benchmark. Baseline results are officially reported by GIFT-Eval (Aksu et al., 2024).

| Type | Pretrained Forecasting Models (Zero-Shot) | | | | | | | | | | Supervised | | Statistical |
| Method | ZEUS (Ours) | Chronos-2 (2025) | TimesFM2.5 (2023) | TiRex (2025) | Xihe$_u$ (2025) | FlowState (2025) | Kairos$_b$ (2025) | Moirai2 (2024) | Toto (2025) | Sundial (2025b) | PatchTST (2023) | DLinear (2023) | Seasonal Naive |
|---|---|---|---|---|---|---|---|---|---|---|---|---|---|
| MASE | **0.693** | 0.698 | 0.705 | 0.716 | 0.701 | 0.726 | 0.742 | 0.728 | 0.750 | 0.750 | 0.849 | 1.061 | 1.000 |
| CRPS | **0.480** | 0.485 | 0.490 | 0.488 | 0.488 | 0.502 | 0.548 | 0.516 | 0.517 | 0.559 | 0.587 | 0.846 | 1.000 |

*Table 3.* Imputation results under `random` and `block` masking, where {12.5%, 25%, 37.5%, 50%} points are masked. Reported metrics are averaged over four mask ratios. Best and second-best results are in **bold** and underlined, respectively. See Table D.3 for full results.

| Models | | Time Series Foundation Models (Zero-shot) | | | | | | | | Task-Specific Models (Supervised) | | | | | | | | | |
| | | ZEUS (Ours) | | MOMENT (2024) | | Timer (2024b) | | UniTS (2024) | | GPT4TS (2023) | | ModernTCN (2024) | | TimesNet (2023) | | PatchTST (2023) | | DLinear (2023) | |
| Metrics | | MSE | MAE | MSE | MAE | MSE | MAE | MSE | MAE | MSE | MAE | MSE | MAE | MSE | MAE | MSE | MAE | MSE | MAE |
|---|---|---|---|---|---|---|---|---|---|---|---|---|---|---|---|---|---|---|---|
| ETTh1 | Random | **0.079** | **0.175** | 0.382 | 0.398 | 0.484 | 0.451 | 0.788 | 0.614 | 0.103 | 0.215 | 0.086 | 0.202 | 0.089 | 0.198 | 0.131 | 0.239 | 0.167 | 0.279 |
| | Block | 0.115 | **0.202** | 0.412 | 0.414 | 0.507 | 0.460 | 0.767 | 0.613 | 0.135 | 0.243 | **0.104** | 0.222 | 0.111 | 0.218 | 0.193 | 0.283 | 0.241 | 0.337 |
| ETTh2 | Random | **0.056** | **0.136** | 0.166 | 0.276 | 0.182 | 0.283 | 0.589 | 0.537 | 0.065 | 0.171 | 0.058 | 0.162 | 0.059 | 0.161 | 0.069 | 0.169 | 0.147 | 0.261 |
| | Block | **0.067** | **0.151** | 0.192 | 0.294 | 0.192 | 0.290 | 0.618 | 0.549 | 0.082 | 0.194 | 0.073 | 0.185 | 0.078 | 0.185 | 0.093 | 0.198 | 0.278 | 0.353 |
| ETTm1 | Random | **0.038** | **0.116** | 0.275 | 0.335 | 0.676 | 0.506 | 0.730 | 0.590 | 0.065 | 0.169 | 0.051 | 0.152 | 0.053 | 0.152 | 0.058 | 0.157 | 0.086 | 0.201 |
| | Block | **0.064** | **0.142** | 0.322 | 0.360 | 0.698 | 0.515 | 0.734 | 0.592 | 0.094 | 0.199 | 0.074 | 0.181 | 0.076 | 0.178 | 0.117 | 0.212 | 0.186 | 0.290 |
| ETTm2 | Random | **0.027** | **0.084** | 0.103 | 0.214 | 0.125 | 0.239 | 0.505 | 0.498 | 0.034 | 0.119 | 0.032 | 0.114 | 0.032 | 0.114 | 0.035 | 0.115 | 0.102 | 0.214 |
| | Block | **0.035** | **0.099** | 0.125 | 0.233 | 0.131 | 0.245 | 0.537 | 0.513 | 0.048 | 0.145 | 0.046 | 0.141 | 0.045 | 0.138 | 0.052 | 0.143 | 0.191 | 0.292 |
| ECL | Random | **0.045** | **0.132** | 0.304 | 0.419 | 0.412 | 0.499 | 0.888 | 0.771 | 0.114 | 0.235 | 0.104 | 0.230 | 0.104 | 0.223 | 0.090 | 0.212 | 0.111 | 0.237 |
| | Block | **0.058** | **0.146** | 0.327 | 0.432 | 0.441 | 0.516 | 0.861 | 0.753 | 0.124 | 0.249 | 0.122 | 0.247 | 0.110 | 0.229 | 0.116 | 0.238 | 0.150 | 0.274 |
| Weather | Random | **0.030** | **0.035** | 0.083 | 0.142 | 0.112 | 0.168 | 0.207 | 0.288 | 0.036 | 0.072 | 0.034 | 0.064 | 0.034 | 0.067 | 0.036 | 0.064 | 0.102 | 0.214 |
| | Block | **0.036** | **0.042** | 0.093 | 0.154 | 0.119 | 0.175 | 0.211 | 0.291 | 0.044 | 0.084 | 0.046 | 0.085 | 0.043 | 0.082 | 0.051 | 0.085 | 0.093 | 0.169 |

ing task. We consider four different prediction horizons {96, 192, 336, 720}, where the best context length is chosen through hyperparameter search. Performance is evaluated using MSE and MAE.

**Results** As shown in Table 1, ZEUS demonstrates strong overall performance among advanced models. Compared with the previous state-of-the-art model, ZEUS achieves an averaged reduction of **9.0%** on MSE and **2.3%** on MAE. Moreover, ZEUS overcomes the common limitation where TSFMs lag behind task-specific models in zero-shot forecasting. When compared to the best-performing TSFM under the zero-shot settings, Timer (Liu et al., 2024b), ZEUS achieves substantial improvements, reducing MSE by **40.3%** and MAE by **25.3%**.

### 4.2. Probabilistic Forecasting

**Setups** Beyond point forecasting, we evaluate our model on the GIFT-Eval benchmark (Aksu et al., 2024) for probabilistic forecasting, which comprises 97 prediction tasks spanning short-, medium-, and long-term forecasting, providing a comprehensive assessment of predictive performance. Following the official protocol, we report MASE and CRPS as evaluation metrics.

**Results** Table 2 reports the aggregated performance across 97 tasks on GIFT-Eval. Among the advanced pretrained

forecasting models, including Chronos-2 (Ansari et al., 2025), TimesFM2.5 (Das et al., 2023), and TiRex (Auer et al., 2025), ZEUS ranks first on both MASE and CRPS, highlighting its strong zero-shot forecasting capabilities.

### 4.3. Imputation

**Setups** Time series imputation aims to fill in missing values of time series and is critical in real-world applications. Following previous work (Liu et al., 2024b; Zhang et al., 2025), we evaluate ZEUS on the ETT, ECL and Weather datasets by masking {12.5%, 25%, 37.5%, 50%} time steps in sequences of length 192. Similar to point forecasting tasks, We use MSE and MAE as evaluation metrics.

Existing TSFMs (Liu et al., 2024b; Goswami et al., 2024; Zhang et al., 2025) are typically evaluated under patchwise missing patterns. However, real-world time series exhibit more diverse missing behaviors, including both pointwise missing and contiguous segment missing with variable lengths (Yu et al., 2025b). To better reflect practical scenarios, we adopt two masking strategies: (1) random masking, which has been widely adopted in prior task-specific models (Wu et al., 2023; donghao & xue, 2024), and (2) block masking, where the lengths of contiguous missing segments are sampled from a Geometry distribution with $p = 0.125$.

**Results** As shown in Table 3, ZEUS consistently achieves superior performance across all datasets and in zero-shot,

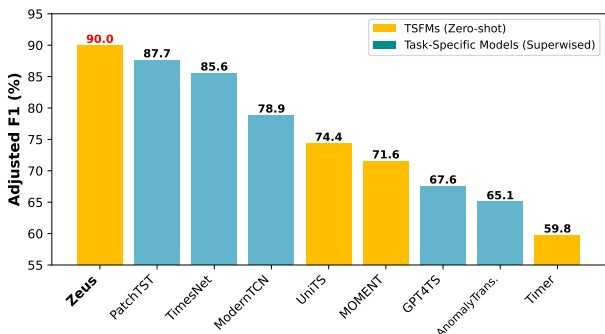

Figure 4. Averaged adjusted F1 score on 42 UCR anomaly detection datasets. See Table 9 for full results.

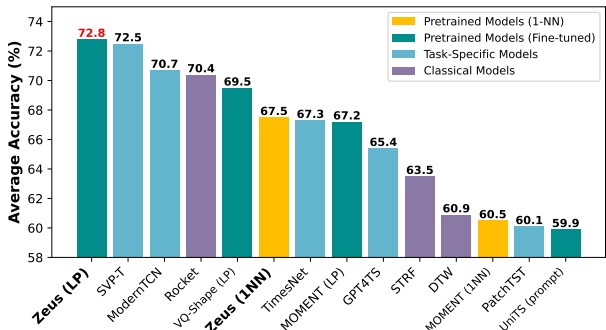

Figure 5. Averaged accuracy on 26 UEA classification datasets, where *LP* denotes linear probing and *prompt* denotes fine-tuning on prompt tokens. See Table 10 for full results.

demonstrating strong robustness to variable-length missing patterns. Patch-based TSFMs (MOMENT, UniTS, and Timer) perform substantially worse than task-specific models, mainly due to the mismatch between their patch-wise pretraining objective and point-level missing patterns at inference, which results in an out-of-distribution scenario. In contrast, ZEUS effectively handles both random and block missing scenarios. Even compared with the strongest task-specific models, ZEUS reduces the averaged MSE by **24.4%** under random masking and **18.8%** under block masking.

### 4.4. Anomaly Detection

**Setups**   To provide a comprehensive evaluation, we conduct experiments on a collection of 42 datasets sourced from the UCR anomaly archive (Wu & Keogh, 2021). This subset that has been adopted in previous studies (Goswami et al., 2024), covering a sufficiently broad range of domains and data sources. We partition the time series into non-overlapping detection windows (Xu et al., 2022) and use the reconstruction error as the criterion for anomaly detection (Wu et al., 2023). The window size is selected via hyperparameter search and the results of all baselines are reported under the optimal settings. We use adjusted F1-score as the evaluation metric (Wu et al., 2023).

**Results**   Figure 4 presents the overall performance of anomaly detection tasks. ZEUS ranks first among all baselines across 42 UCR datasets. Under the zero-shot setting, ZEUS outperforms advanced task-specific models trained in full-shot. Moreover, ZEUS achieves a **21.0%** improvement in F1-score over UniTS (Gao et al., 2024), the second-best performing TSFM on this benchmark.

### 4.5. Classification

**Setups**   We conduct evaluations on 26 UEA classification datasets (Bagnall et al., 2018). Prior TSFMs relies on fine-tuning (Gao et al., 2024; Zhang et al., 2025) or linear

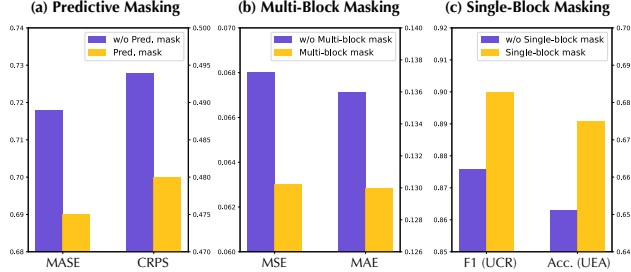

Figure 6. Ablation results of ZEUS.

probing (Goswami et al., 2024) for classification tasks. In our evaluation, ZEUS is primarily assessed in a tuning-free setting using a non-parametric 1-nearest neighbor (1-NN) classifier for evaluation, with optional PCA whitening applied for feature normalization. In addition, we also report linear probing results to assess the linear separability of the learned representations and to facilitate comparison with existing methods. Following prior work (Goswami et al., 2024), we use the accuracy as the evaluation metric.

**Results**   As shown in Figure 5, ZEUS achieves superior performance across the 26 UEA classification datasets under different evaluation protocols. In particular, ZEUS with linear probing attains the highest averaged accuracy among all compared methods, outperforming both task-specific models and other pretrained baselines. Notably, ZEUS with 1-NN evaluation also delivers competitive performance, being on par with advanced models such as TimesNet and MOMENT-LP, while outperforming MOMENT under the same 1-NN setting by a clear margin of 7.0 percentage points. Overall, these results highlight the effectiveness and generalizability of ZEUS's pretrained representations.

### 4.6. Ablation Study

In this section, we investigate the effectiveness of our proposed MOTM by ablating individual masking strategies.

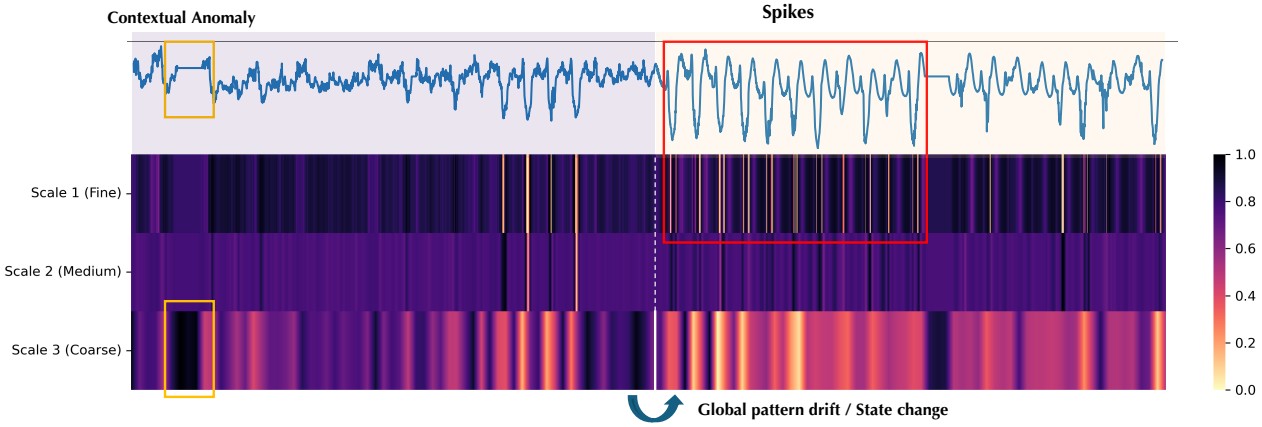

*Figure 7.* Multi-scale feature-norm heatmaps, which illustrates the roles of different scales: fine-scale representations are sensitive to local variations and extreme values (red boxes), mid-scale stripe patterns capture intrinsic periodicity, and coarse-scale representations model global pattern shifts (white vertical line) and contextual anomalies (yellow boxes).

When a specific mask is removed, we keep the expected masking ratio fixed by proportionally reallocating it to the other masks. As illustrated in Figure 6 (a), removing the predictive mask leads to a clear performance drop on the GiftEval benchmark, validating its crucial role in enhancing extrapolation capabilities and forecasting precision. Similarly, Figure 6 (b) shows the absence of multi-block mask degrades ZEUS's performance on imputation tasks, indicating its importance for interpolation and local pattern reconstruction. Furthermore, the removal of the single-block mask leads to consistent performance degradation in both anomaly detection and classification, as shown in Figure 6 (c), suggesting that single-block reconstruction guides the model to capture global consistency.

### 4.7. Model Analysis

**Representation Analysis**   We conduct a representation analysis to illustrate the roles of different temporal scales in ZEUS. Specifically, we select a sample sequence from the ETTm1 dataset and visualize the feature-norm heatmaps of the representations at multiple scales, as shown in Figure 7.

It can be observed that fine-scale representations are highly sensitive to local variations and extreme values. The regions highlighted by red boxes indicate that these representations effectively capture each negative spike in the sequence. In contrast, mid-scale representations are less responsive to local fluctuations; instead, their stripe-like patterns reflect the underlying periodic structures. Coarse-scale representations clearly characterize global pattern changes in the sequence. For instance, the white vertical line marks the transition from small-amplitude fluctuations to periodic negative impulses, which is reflected by a color change in the heatmap from purple to yellow. Moreover, the regions highlighted by

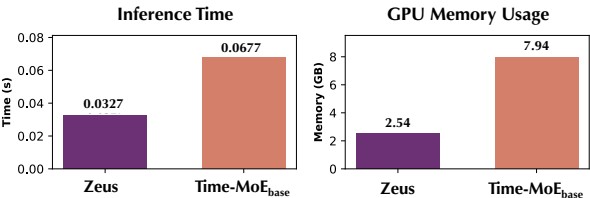

*Figure 8.* Efficiency comparison between ZEUS and Time-MoE$_{base}$, two point-tokenized models with comparable model size. Results are averaged over 1,000 runs on sequences of length $L=4096$.

yellow boxes demonstrate that large-scale representations are effective in capturing contextual anomalies.

**Efficiency Analysis**   Conventional point-tokenized Transformers suffers from the high computational cost, as processing a sequence of length $L$ with dimension $d$ over $N$ layers incurs $\mathcal{O}(NL^2d)$ computations due to full-resolution self-attention. In contrast, ZEUS performs most attention computations at coarser temporal resolutions, resulting in a reduced complexity of

$$\mathcal{C}_{\text{ZEUS}} = \sum_i \mathcal{O}\left(N_i \left(\frac{L}{s_i}\right)^2 d_i\right), \qquad (6)$$

where $N_i$ and $d_i$ denote the number of layers and the hidden dimension at scale $s_i$. Under our model configuration (Appendix A), ZEUS achieves a $3.8\times$ reduction in self-attention FLOPs compared to a vanilla Transformer with the same depth (Appendix C.2). Additionally, we conduct an empirical efficiency comparison between ZEUS and Time-MoE$_{base}$ (Shi et al., 2025), a point-tokenized pretrained model with approximately 113M parameters. Both models are evaluated

under the same environment with FlashAttention enabled, ensuring a fair comparison. As shown in Figure 8, ZEUS is $2.1\times$ faster and $3.1\times$ memory efficient than Time-MoE$_{\text{base}}$, demonstrating its scalability over long context.

## 5. Conclusion

In this work, we propose ZEUS, a unified multi-scale Transformer with point-wise tokenization and a U-shaped hierarchy that effectively balances fine-grained temporal fidelity with long-sequence efficiency. Complementing the architectural design, we introduce MOTM, a multi-objective temporal masking strategy that jointly supports extrapolation, interpolation, and global abstraction within a single pretraining framework. Moreover, we construct Aegis-Syn, a synthetic dataset that extends KernelSynth with richer temporal patterns. Extensive experiments across five representative downstream tasks demonstrate that ZEUS achieves strong and consistent performance in a fully tuning-free setting, highlighting its potential as a truly general-purpose TSFM. We believe this work takes a meaningful step toward versatile and scalable foundation models for time series analysis. Future work will explore multivariate modeling and extend the proposed framework to a broader range of downstream tasks. See Appendix F for detailed discussion on limitations and future directions.

## Acknowledgments

This work is supported by the NSFC underGrant Nos.62372430 and 62502505, the Youth Innovation Promotion Association CAS No.2023112, the Postdoctoral Fellowship Program of CPSF under Grant Number GZC20251078, the China Postdoctoral Science Foundation No.2025M77154 and HUA Innovation fundings. We sincerely thank all the anonymous reviewers who gerenously contributed their time and efforts.

## Impact Statement

This paper presents work whose goal is to advance the field of Machine Learning. There are many potential societal consequences of our work, none which we feel must be specifically highlighted here.

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

# A. Implementation Details

## A.1. Model Configuration

ZEUS has approximately 100M parameters, and the detailed model configurations are summarized in Table 4. We set the maximum context length to 4096. By using FlashAttention (Dao, 2024) and preventing padded tokens from participating in attention, we ensure that the additional padding does not incur significant computational overhead. The model are trained for 200,000 steps with a global batch size of 512. We employ the AdamW optimizer (Kingma & Ba, 2014) with a cosine learning rate schedule and a warmup phase of 10,000 steps. The initial learning rate is set to $1 \times 10^{-3}$. All experiments are implemented using PyTorch (Paszke et al., 2019) and trained on 4 NVIDIA H100 GPUs.

*Table 4.* Hyperparameters of ZEUS.

| Hyperparameter | Value |
|---|---|
| Scales | $[1, 8, 32, 8, 1]$ |
| # Layers | $[1, 3, 3, 3, 2]$ |
| Hidden size | $[384, 768, 768, 768, 384]$ |
| # Heads | $[6, 12, 12, 12, 6]$ |
| Intermediate size | $[1536, 3072, 3072, 3072, 1536]$ |
| # Parameters | 100M |

## A.2. Unified Downstream Task Formulation

This section describes how ZEUS is applied to various downstream tasks.

**Forecasting**   Given a historical time series context $\mathbf{x} \in \mathbb{R}^{T \times C}$, where $T$ denotes the context length and $C$ the number of variables, ZEUS performs forecasting by formulating the task as a masked sequence completion problem. ZEUS adopt channel independent strategy that treat $\mathbf{x}$ as $C$ univariate time series.

To perform forecasting on a horizon $H$, the context is concatenated with $H$ [MASK] tokens, i.e., $\tilde{\mathbf{x}} = [\mathbf{x}, \mathbf{M}] \in \mathbb{R}^{(T+H) \times C}$, and then fed to ZEUS. The output of ZEUS is a quantile reconstruction tensor $\mathbf{z} \in \mathbb{R}^{(T+H) \times C \times |Q|}$, where $|Q|$ denotes the number of quantile levels. We retain the predictions at the final $H$ steps as the probabilistic forecast $\mathbf{y} \in \mathbb{R}^{H \times C \times |Q|}$, and the point forecast is obtained by averaging predictions across all quantiles.

**Imputation**   For the imputation task, ZEUS follows the same reconstruction paradigm as used during pretraining. Missing values are replaced with [MASK] tokens and fed into ZEUS, which produces quantile predictions for all time steps. The outputs at masked positions are used as imputed values, with point estimates obtained by averaging across quantiles.

**Anomaly Detection**   ZEUS supports anomaly detection under two complementary paradigms: *prediction-based* and *reconstruction-based*. The prediction-based approach detects anomalies by forecasting future values and measuring prediction errors, which is well suited for streaming scenarios where only past observations are available. In contrast, the reconstruction-based approach masks a target segment and reconstructs it using both past and future context, enabling more accurate detection by exploiting full contextual information. In this work, we provide a formal description of the reconstruction-based formulation, while the prediction-based variant follows an analogous procedure.

Given a historical time series $\mathbf{X}_{1:L} \in \mathbb{R}^{L \times C}$ and a target window size $T < L$, we partition the sequence into non-overlapping subsequences using a sliding window with stride $T$. For each target segment $\mathbf{X}_{t:t+T}$, we construct the input by masking the target window and concatenating it with its surrounding context, forming $\tilde{\mathbf{X}} = [\mathbf{X}_{t-W:t}, \mathbf{M}, \mathbf{X}_{t+T:t+T+W}]$, where $W$ denotes the context length and $\mathbf{M}$ denotes $T$ consecutive [MASK] tokens. The resulting sequence is then fed into ZEUS to reconstruct the masked segment.

The model outputs quantile predictions $\mathbf{z} \in \mathbb{R}^{T \times C \times |Q|}$ for the target window. An anomaly score is computed based on the reconstruction error, with higher errors indicating a higher likelihood of anomalies. By default, we use MAE as the reconstruction error for anomaly scoring. However, for sequences containing impulsive patterns, we selectively adopt relative MAE as the error metric, which normalizes the absolute error by the signal magnitude. This choice prevents large but expected impulses from being erroneously identified as anomalies.

**Classification**   ZEUS performs classification using a 1-nearest neighbor (1-NN) classifier on the learned representations. Specifically, given the train dataset $\mathcal{D}_{train}$ and a sample, ZEUS predict label $\hat{y}$ with its hidden states $\mathbf{h} \in \mathbb{R}^{T \times C \times d}$ by

$$\mathbf{z} = \text{Flatten}\left(\text{GlobalPool}(\mathbf{h})\right), \tag{7}$$

$$i^* = \arg \max_{i \in \mathcal{D}_{\text{train}}} \text{sim}(\mathbf{z}, \mathbf{z}_i), \tag{8}$$

$$\hat{y} = y_{i^*}, \tag{9}$$

where $\text{GlobalPool}(\cdot)$ aggregates the temporal dimension, for which we default to max pooling, $\text{Flatten}(\cdot)$ flattens the channel dimension, and $\text{sim}(\cdot, \cdot)$ denotes the cosine similarity. $\mathbf{h}$ can be taken from a specific scale or formed by concatenating representations from multiple scales. In practice, we typically use the representation from the penultimate scale ($s_4 = 8$) or the coarsest scale ($s_3 = 32$).

As an alternative to 1-NN classification, we also consider a linear probe applied on top of $\mathbf{z}$, where the backbone of ZEUS is frozen during training.

### A.3. Technical Details of MOTM

This section provides the technical details of MOTM that is used to pretrain ZEUS.

**Masking Ratio**   The overall masking ratio is sampled from a uniform distribution, i.e., $p \sim \mathcal{U}(0, 0.5)$, with the expected rate being $0.25$. A low masking ratio is used to simulate short-term forecasting scenarios, where the model need to accurately infer future dynamics from long historical context. In contrast, a high masking ratio poses a more challenging setting and is used to emulate imputation scenarios with severe missingness. Compared to a fixed masking ratio that is typically adopted by prior work (Goswami et al., 2024; Zhang et al., 2025), a variable masking ratio prevents the model from overfitting to a single missingness level and improves robustness across diverse missing rates.

**Temporal Range**   To increase the diversity of contexts observed by the model, we sample variable sequence lengths and randomly crop the input sequences, padding them to a maximum context length of 4096. Specifically, the sequence length is sampled from a piecewise uniform distribution: with probability 0.2 from $[64, 512]$, 0.2 from $[513, 2048]$, and 0.6 from $[2049, 4096]$. This design ensures that short- and mid-range temporal dependencies are sufficiently observed, while most sequences fall into the long-range regime and require minimal padding, thereby balancing training efficiency.

## B. Related Work

### B.1. Systematic Comparison with Closely Related Work

In this section, we present a systematic comparison with closely related TSFMs. In this work, we use the term *time series foundation models* to denote pretrained models that are capable of supporting a wide range of downstream time series analysis tasks, in contrast to *pretrained forecasting models* that are designed exclusively for forecasting. In addition, some LLM-based approaches, such as GPT4TS (Zhou et al., 2023), do not involve pretraining on time series data. Existing studies further indicate that the performance gains do not primarily stem from the LLM backbone (Tan et al., 2024). Therefore, we classify these methods as task-specific models. While state-of-the-art pretrained forecasting models are included in the forecasting evaluations, this section focuses on comparisons among TSFMs.

Table 5 summarizes the key differences between ZEUS and representative TSFMs, including TimesBERT (Zhang et al., 2025), MOMENT (Goswami et al., 2024), UniTS (Gao et al., 2024), and Timer (Liu et al., 2024b), in terms of model architecture, pretraining scale, tokenization strategy, and downstream task support. Existing TSFMs predominantly adopt patch-based tokenization, whereas ZEUS employs point-wise tokenization to preserve fine-grained details. In addition, ZEUS is pretrained at a larger scale and with a substantially longer context length than prior TSFMs. As TimesBERT is not publicly available, it is not included as a baseline in our experiments.

Beyond these differences, a more salient distinction lies in downstream task coverage. ZEUS supports a substantially broader set of time series analysis tasks than existing TSFMs, enabling forecasting, imputation, anomaly detection, and classification in a tuning-free manner. In contrast, existing TSFMs typically support only a subset of downstream tasks

*Table 5.* Comparison with existing TSFMs. A dash (−) indicates that the information is unspecified or not explicitly reported. For downstream task support, ✓ denotes zero-shot capability, ✗ indicates unsupported task, and ◯ indicates that fine-tuning is required.

| | **ZEUS (Ours)** | **TimesBERT (2025)** | **MOMENT (2024)** | **UniTS (2024)** | **Timer (2024b)** |
|---|---|---|---|---|---|
| Architecture | Encoder-only | Encoder-only | Encoder-only | Encoder-only | Decoder-only |
| Model Size | 100M | 86M | 40M, 125M, 385M | 8M | 29M, 50M, 67M |
| Pretraining Scale | 300B | 260B | 1B | 35M | 28B |
| Context Length | 4096 | - | 512 | - | 1440 |
| Tokenization | Point | Patch | Patch | Patch | Patch |
| Open-sourced | ✓ | ✗ | ✓ | ✓ | ✓ |
| **Supported Downstream Tasks** | | | | | |
| Long-term Forecasting | ✓ | ✗ | ◯ | ◯ | ✓ |
| Short-term Forecasting | ✓ | ◯ | ✓ | ◯ | ✓ |
| Probabilistic Forecasting | ✓ | ✗ | ✗ | ✗ | ✗ |
| Imputation | ✓ | ◯ | ✓ | ◯ | ◯ |
| Anomaly Detection | ✓ | ◯ | ✓ | ◯ | ◯ |
| Classification | ✓ | ◯ | ◯ | ◯ | ✗ |

or rely on task-specific fine-tuning or adaptations. Overall, this comparison positions ZEUS as a more general-purpose foundation model with comprehensive and unified multi-task capabilities.

### B.2. Multi-Scale Architectures

Multi-scale modeling has been widely explored in task-specific models (Zhang & Yan, 2023; Yu et al., 2023; 2025a; Deng et al., 2024; Shao et al., 2025b), where hierarchical representations are adopted to capture temporal dependencies at different resolutions. These studies have demonstrated the effectiveness of multi-scale designs for improving model performance. However, despite their success in task-specific settings, most existing TSFMs still rely on single-scale architectures. Only a few exceptions explored multi-scale structures, and ZEUS differs from these works fundamentally in both information flow and architectural design. For instance, TTM (Ekambaram et al., 2024) and Xihe (Sun et al., 2025) follows a fine-to-coarse paradigm, progressively aggregating local patterns, while Kairos (Feng et al., 2025) adopts a coarse-to-fine mixture-of-size patching strategy. In contrast, ZEUS is the first to introduce a fine-to-coarse-to-fine multi-scale architecture, which preserves point-level details while simultaneously enhancing long-sequence scalability.

### B.3. Time Series Tokenization

Time series tokenization has been extensively studied in recent years. Beyond the widely adopted point tokens (Zhou et al., 2021; Deng et al., 2022) and patch tokens (Nie et al., 2023; Tang & Zhang, 2025), prior works have also explored alternative tokenization paradigms such as variable-wise (Shao et al., 2022; Liu et al., 2024a), frequency-wise (Yi et al., 2023), phase-wise tokenization (Niu et al., 2025), and various improvements to patch-wise tokenization (Huang et al., 2024; Abeywickrama et al., 2026). However, in the context of TSFMs, the design space of tokenization is constrained by the requirement to support arbitrary input lengths. As a result, many existing tokenization improvements are difficult to directly apply to TSFMs. In addition, point-wise tokenization is less favored due to its limited scalability under long-context settings. More recently, tokenization strategies specifically designed for TSFMs have been proposed, including domain-specific tokenization (Özgün Turgut et al., 2025) and wavelet-based tokenization (Masserano et al., 2025). Nevertheless, patch-wise tokenization remains the dominant paradigm in current TSFM architectures.

## C. Supplementary Analysis

### C.1. Limitations of Patch Tokenization

In this section, we further analyze the drawbacks of patch tokenization and provide empirical evidence to substantiate our claims.

Most existing pretrained time series models adopt patch tokenization. Although patching effectively increases the semantic density of tokens and reduces the total number of tokens (Nie et al., 2023), it also introduces several nontrivial limitations.

First, patching inherently entangles fine-grained temporal variations within a patch, which weakens the model's ability to reason at the point level. This mismatch becomes particularly evident in tasks that require precise temporal localization, such as point-wise imputation and anomaly detection. Moreover, models pretrained with patch-level reconstruction objectives tend to overfit to patch-wise missing patterns, leading to degraded robustness when the corruption pattern shifts to point-level missingness. We provide empirical evidence for these limitations in Table 6. Notably, although point-wise missing constitutes a strictly simpler corruption pattern than patch-wise missing, the performance of MOMENT drops by over 20% when evaluated under point-missing imputation. Last but not least, patch tokenization struggles to capture high-frequency dynamics. Consider a sequence whose period equals the patch length: all patch tokens become identical, causing the Transformer to degenerate into a feed-forward MLP.

*Table 6.* Comparison of MOMENT's performance between patch-level missing and point-level missing. Results are averaged over four masking ratios $\{12.5\%, 25\%, 37.5\%, 50\%\}$.

| Datasets | | ETTm1 | | ETTh2 | | Weather | | Avg | |
|---|---|---|---|---|---|---|---|---|---|
| Metric | | MSE | MAE | MSE | MAE | MSE | MAE | MSE | MAE |
| MOMENT | Patch | 0.226 | 0.284 | 0.133 | 0.239 | 0.071 | 0.102 | 0.143 | 0.208 |
| | Point | 0.275 | 0.335 | 0.166 | 0.276 | 0.083 | 0.142 | 0.175 | 0.251 |
| | $\Delta$ | -21.7% | -18.0% | -24.8% | -15.5% | -16.9% | -28.2% | -22.4% | -20.7% |

### C.2. Detailed Efficiency Analysis

This section provides a detailed comparison of the computational complexity between ZEUS and a vanilla Transformer.

**Baseline: Vanilla Transformer**  We consider a vanilla Transformer with point-wise tokenization and the same total number of layers as ZEUS. For a sequence of length $L$ and hidden dimension $d$, the self-attention module incurs a computational cost of $\mathcal{O}(BL^2d)$ per layer, which dominates the overall FLOPs. With $N$ layers, the total attention complexity is

$$\mathcal{C}_{\text{vanilla}} = \mathcal{O}(BNL^2d). \tag{10}$$

**ZEUS**  ZEUS employs a multi-scale architecture with temporal scales $\{s_i\}$. The attention complexity at scale $s_i$ is computed as Eq.(6).

**Quantitative Comparison**  Using the configuration in our implementation, most layers in ZEUS operate on temporally downsampled representations (e.g., $L/8$ and $L/32$), while only a small number of layers attend at full resolution. Let $d = 384$ and $N = 12$, then we obtain

$$\mathcal{C}_{\text{ZEUS}} = B \cdot L^2 \Big( 3 \cdot 384 + \frac{6}{8^2} \cdot 768 + \frac{3}{32^2} \cdot 768 \Big), \tag{11}$$

$$\mathcal{C}_{\text{vanilla}} = B \cdot L^2 \cdot 12 \cdot 384, \tag{12}$$

$$\mathcal{C}_{\text{ZEUS}} \approx 0.27\,\mathcal{C}_{\text{vanilla}}, \tag{13}$$

which indicates that ZEUS achieves an approximately $3.8\times$ reduction in self-attention FLOPs compared to a vanilla Transformer with the same depth.

## D. Experimental Details

### D.1. Point Forecasting

**Benchmarks**  We conduct experiments on six widely-recognized datasets in long-term forecasting benchmarks (Wu et al., 2023; Shao et al., 2024), including:

- **ETT** (Electricity Transformer Temperature) (Zhou et al., 2021) contains 7 features of electricity transformer data collected from two separate counties from July 2016 to July 2018. It contains four datasets: **ETTh1, ETTh2, ETTm1, ETTm2**, where ETTh1 and ETTh2 are recorded every hour, and ETTm1 and ETTm2 are recorded every 15 minutes.

- **ECL** (Electricity) (Wu et al., 2021) records the hourly electricity consumption data of 321 clients from 2012 to 2014. Each variable represents a client's electricity consumption.

- **Weather**(Wu et al., 2021) includes 21 meteorological factors collected every 10 minutes from the weather station of the Max Planck Biogeochemistry Institute in 2020.

Following common practice, we consider four prediction horizons $\{96, 192, 336, 720\}$. For each horizon, the context length is selected via hyperparameter search from the set $\{512, 720, 1024, 2048, 3072\}$.

**Baselines**   We compare our methods with three categories of baselines: (1) *TSFMs*: **MOMENT** (Goswami et al., 2024), **Timer** (Liu et al., 2024b), **UniTS** (Gao et al., 2024); (2) *Pretrained forecasting models*: **Kairos** (Feng et al., 2025), **Toto** (Cohen et al., 2025), **Sundial** (Liu et al., 2025b); (3) *Task-specific models*: **ModernTCN** (donghao & xue, 2024), **GPT4TS** (Zhou et al., 2023), **TimesNet** (Wu et al., 2023), **PatchTST** (Nie et al., 2023).

For baselines that include multiple model variants, we report results for the variant with the strongest overall performance (typically the largest ones). Subscripts $_b$ and $_l$ denote the base and large variants, respectively. Notably, the prediction head of MOMENT is randomly initialized and therefore requires fine-tuning before being applicable to forecasting, making it unsuitable for zero-shot evaluation. Consequently, we use its reconstruction head instead and formulate the forecasting task as an extrapolative reconstruction task, following the same setup used for ZEUS.

**Metrics**   We adopt **Mean Squared Error (MSE)** and **Mean Absolute Error (MAE)** as evaluation metrics, where lower values indicate better performance. They are defined as follows.

$$\text{MSE}(\hat{y}, y) = \frac{1}{N \cdot T} \sum_{i=1}^{N} \sum_{t=1}^{T} \left( \hat{y}_t^i - y_t^i \right)^2, \tag{14}$$

$$\text{MAE}(\hat{y}, y) = \frac{1}{N \cdot T} \sum_{i=1}^{N} \sum_{t=1}^{T} |\hat{y}_t^i - y_t^i|. \tag{15}$$

**Full Results**   The full results of point forecasting evaluation are provided in Table 7.

### D.2. Probabilistic Forecasting

**Benchmarks**   For probabilistic forecasting, we evaluate ZEUS's performance on the **GIFT-Eval** benchmark (Aksu et al., 2024). GIFT-Eval comprises 23 datasets spanning multiple domains, including nature, energy, healthcare, finance, transportation, and cloud operations, covering 144,000 time series with 177 million data points. These datasets define a total of 97 forecasting configurations, including 55 short-term, 21 medium-term, and 21 long-term forecasting tasks, thereby providing a comprehensive assessment of the model's predictive capabilities. Compared to long-term forecasting benchmarks used in point forecasting evaluation, GIFT-Eval places greater emphasis on short-term forecasting and additionally provides evaluation metrics for probabilistic predictions.

**Baselines**   We include baselines: **Chronos-2** (Ansari et al., 2025), **TimesFM2.5** (Das et al., 2023), **TiRex** (Auer et al., 2025), **Xihe**$_{ultra}$ (Sun et al., 2025), **FlowState** (Graf et al., 2025), **Kairos** (Feng et al., 2025), **Moirai2** (Woo et al., 2024), **Toto** (Cohen et al., 2025), **Sundial** (Liu et al., 2025b). The performance of baselines is taken from the official benchmark reports, using results available up to December 31, 2025.

**Metrics**   Following the official protocol of GIFT-Eval, we report **Mean Absolute Scaled Error (MASE)** and **Continuous Ranked Probability Score (CRPS)** as evaluation metrics. Both metrics are lower-is-better.

**MASE** (Hyndman & Koehler, 2006) evaluates point forecasting performance by scaling the mean absolute error (MAE) of a model against that of a seasonal naive benchmark. It is defined as

$$\text{MASE} = \frac{\text{MAE}_{\text{model}}}{\text{MAE}_{\text{seasonal naive}}} \tag{16}$$

*Table 7.* Zero-shot point forecasting results across four prediction lengths {96, 192, 336, 720}. Datasets in pre-training are not evaluated on corresponding models, which are denoted by the dash (−). Best and second-best results are shown in **bold** and underlined, respectively.

| Models | | Time Series Foundation Models (Zero-shot) | | | | | | | | Pretrained Forecasting Models (Zero-shot) | | | | | | | | | | Task-Specific Models (Full-shot) | | | | | | | |
|---|---|---|---|---|---|---|---|---|---|---|---|---|---|---|---|---|---|---|---|---|---|---|---|---|---|---|---|
| | | **ZEUS** (Ours) | | MOMENT (2024) | | Timer (2024b) | | UniTS (2024) | | Kairos$_b$ (2025) | | Toto$_b$ (2025) | | Sundial$_l$ (2025b) | | Time-MoE$_l$ (2025) | | ChronosBolt$_b$ (2024) | | ModernTCN (2024) | | GPT4TS (2023) | | TimesNet (2023) | | PatchTST (2023) | |
| Metric | | MSE | MAE | MSE | MAE | MSE | MAE | MSE | MAE | MSE | MAE | MSE | MAE | MSE | MAE | MSE | MAE | MSE | MAE | MSE | MAE | MSE | MAE | MSE | MAE | MSE | MAE |
| ETTh1 | 96 | **0.341** | **0.372** | 0.662 | 0.548 | 0.437 | 0.425 | 0.423 | 0.437 | 0.396 | 0.385 | 0.382 | 0.381 | 0.346 | 0.383 | 0.350 | 0.382 | 0.420 | 0.388 | 0.381 | 0.401 | 0.388 | 0.404 | 0.408 | 0.426 | 0.382 | 0.403 |
| | 192 | **0.371** | **0.393** | 0.697 | 0.567 | 0.503 | 0.464 | 0.439 | 0.448 | 0.434 | 0.408 | 0.428 | 0.408 | 0.386 | 0.410 | 0.388 | 0.412 | 0.486 | 0.424 | 0.419 | 0.425 | 0.434 | 0.427 | 0.456 | 0.451 | 0.416 | 0.426 |
| | 336 | **0.384** | **0.403** | 0.700 | 0.574 | 0.510 | 0.470 | 0.460 | 0.452 | 0.452 | 0.418 | 0.457 | 0.422 | 0.410 | 0.426 | 0.411 | 0.430 | 0.517 | 0.445 | 0.443 | 0.441 | 0.463 | 0.446 | 0.508 | 0.496 | 0.443 | 0.444 |
| | 720 | **0.412** | **0.426** | 0.802 | 0.629 | 0.545 | 0.494 | 0.662 | 0.575 | 0.426 | 0.427 | 0.472 | 0.440 | 0.438 | 0.459 | 0.427 | 0.455 | 0.493 | 0.457 | 0.434 | 0.459 | 0.465 | 0.469 | 0.567 | 0.521 | 0.466 | 0.474 |
| | Avg. | **0.377** | **0.399** | 0.715 | 0.580 | 0.499 | 0.463 | 0.496 | 0.478 | 0.427 | 0.410 | 0.435 | 0.413 | 0.395 | 0.420 | 0.394 | 0.420 | 0.479 | 0.429 | 0.419 | 0.432 | 0.438 | 0.437 | 0.485 | 0.469 | 0.427 | 0.437 |
| ETTh2 | 96 | 0.269 | 0.318 | 0.343 | 0.396 | 0.315 | 0.351 | 0.349 | 0.376 | 0.276 | 0.319 | 0.273 | 0.310 | 0.269 | 0.330 | 0.302 | 0.354 | **0.255** | **0.305** | 0.276 | 0.341 | 0.325 | 0.373 | 0.315 | 0.359 | 0.280 | 0.349 |
| | 192 | **0.319** | 0.358 | 0.372 | 0.414 | 0.428 | 0.417 | 0.423 | 0.420 | 0.353 | 0.369 | 0.339 | 0.356 | 0.325 | 0.373 | 0.364 | 0.385 | 0.333 | **0.355** | 0.340 | 0.384 | 0.402 | 0.425 | 0.435 | 0.422 | 0.351 | 0.393 |
| | 336 | **0.341** | **0.381** | 0.396 | 0.429 | 0.463 | 0.450 | 0.459 | 0.455 | 0.385 | 0.394 | 0.374 | 0.387 | 0.354 | 0.400 | 0.417 | 0.425 | 0.381 | 0.389 | 0.365 | 0.406 | 0.429 | 0.455 | 0.481 | 0.456 | 0.388 | 0.420 |
| | 720 | **0.352** | **0.398** | 0.464 | 0.473 | 0.447 | 0.457 | 0.478 | 0.472 | 0.386 | 0.412 | 0.375 | 0.400 | 0.389 | 0.443 | 0.537 | 0.496 | 0.393 | 0.408 | 0.401 | 0.435 | 0.464 | 0.480 | 0.455 | 0.461 | 0.424 | 0.445 |
| | Avg. | **0.320** | 0.364 | 0.394 | 0.428 | 0.413 | 0.419 | 0.427 | 0.431 | 0.350 | 0.374 | 0.340 | **0.363** | 0.334 | 0.387 | 0.405 | 0.415 | 0.341 | 0.364 | 0.346 | 0.392 | 0.405 | 0.433 | 0.422 | 0.425 | 0.361 | 0.402 |
| ETTm1 | 96 | **0.272** | 0.324 | 0.610 | 0.517 | 0.690 | 0.526 | 0.643 | 0.507 | 0.295 | 0.322 | 0.320 | 0.333 | 0.273 | 0.329 | 0.309 | 0.357 | 0.303 | **0.311** | 0.283 | 0.340 | 0.298 | 0.349 | 0.358 | 0.373 | 0.278 | 0.330 |
| | 192 | **0.309** | **0.349** | 0.692 | 0.549 | 0.745 | 0.560 | 0.688 | 0.532 | 0.336 | 0.355 | 0.371 | 0.364 | 0.312 | 0.357 | 0.346 | 0.381 | 0.370 | 0.352 | 0.326 | 0.363 | 0.338 | 0.371 | 0.428 | 0.412 | 0.327 | 0.361 |
| | 336 | **0.334** | **0.368** | 0.753 | 0.567 | 0.949 | 0.637 | 0.704 | 0.547 | 0.367 | 0.380 | 0.408 | 0.388 | 0.343 | 0.378 | 0.373 | 0.408 | 0.410 | 0.382 | 0.363 | 0.384 | 0.370 | 0.391 | 0.494 | 0.447 | 0.363 | 0.388 |
| | 720 | **0.372** | **0.396** | 0.799 | 0.584 | 0.965 | 0.650 | 0.723 | 0.565 | 0.393 | 0.402 | 0.485 | 0.426 | 0.397 | 0.413 | 0.475 | 0.477 | 0.497 | 0.428 | 0.425 | 0.418 | 0.421 | 0.421 | 0.550 | 0.480 | 0.414 | 0.423 |
| | Avg. | **0.322** | **0.359** | 0.714 | 0.554 | 0.837 | 0.593 | 0.690 | 0.538 | 0.348 | 0.365 | 0.378 | 0.396 | 0.331 | 0.369 | 0.376 | 0.405 | 0.395 | 0.368 | 0.346 | 0.376 | 0.357 | 0.383 | 0.458 | 0.428 | 0.346 | 0.376 |
| ETTm2 | 96 | 0.168 | 0.247 | 0.276 | 0.342 | 0.213 | 0.295 | 0.235 | 0.306 | **0.160** | 0.238 | 0.172 | 0.237 | 0.172 | 0.255 | 0.197 | 0.286 | 0.164 | **0.232** | 0.172 | 0.266 | 0.171 | 0.262 | 0.175 | 0.255 | 0.162 | 0.249 |
| | 192 | 0.222 | 0.287 | 0.322 | 0.368 | 0.306 | 0.348 | 0.291 | 0.338 | **0.220** | 0.283 | 0.232 | **0.280** | 0.227 | 0.296 | 0.250 | 0.322 | 0.233 | **0.280** | 0.226 | 0.299 | 0.244 | 0.310 | 0.251 | 0.305 | 0.221 | 0.291 |
| | 336 | **0.268** | 0.319 | 0.384 | 0.401 | 0.433 | 0.428 | 0.346 | 0.372 | 0.271 | **0.317** | 0.290 | 0.320 | 0.275 | 0.331 | 0.337 | 0.375 | 0.299 | 0.323 | 0.281 | 0.333 | 0.293 | 0.346 | 0.307 | 0.343 | 0.270 | 0.323 |
| | 720 | **0.336** | **0.366** | 0.453 | 0.441 | 0.539 | 0.482 | 0.441 | 0.433 | 0.357 | 0.373 | 0.372 | 0.375 | 0.343 | 0.378 | 0.480 | 0.461 | 0.414 | 0.391 | 0.380 | 0.391 | 0.391 | 0.407 | 0.412 | 0.407 | 0.369 | 0.386 |
| | Avg. | **0.249** | 0.305 | 0.359 | 0.388 | 0.373 | 0.388 | 0.328 | 0.362 | 0.252 | **0.303** | 0.267 | **0.303** | 0.254 | 0.315 | 0.258 | 0.315 | 0.278 | 0.307 | 0.265 | 0.322 | 0.275 | 0.331 | 0.286 | 0.328 | 0.256 | 0.312 |
| ECL | 96 | **0.126** | 0.213 | 0.734 | 0.685 | 0.193 | 0.232 | 0.349 | 0.417 | - | - | 0.129 | **0.213** | 0.130 | 0.227 | - | - | - | - | 0.131 | 0.227 | 0.136 | 0.233 | 0.184 | 0.289 | 0.139 | 0.235 |
| | 192 | **0.144** | 0.230 | 0.811 | 0.733 | 0.316 | 0.364 | 0.361 | 0.429 | - | - | 0.145 | **0.229** | 0.150 | 0.247 | - | - | - | - | 0.146 | 0.243 | 0.154 | 0.251 | 0.192 | 0.295 | 0.153 | 0.248 |
| | 336 | **0.161** | 0.248 | 0.859 | 0.755 | 0.342 | 0.383 | 0.430 | 0.483 | - | - | 0.163 | **0.247** | 0.170 | 0.268 | - | - | - | - | 0.166 | 0.264 | 0.172 | 0.270 | 0.193 | 0.299 | 0.168 | 0.267 |
| | 720 | 0.197 | **0.280** | 1.194 | 0.876 | 0.366 | 0.405 | 0.654 | 0.632 | - | - | 0.206 | 0.282 | 0.214 | 0.307 | - | - | - | - | **0.194** | 0.288 | 0.208 | 0.298 | 0.222 | 0.320 | 0.208 | 0.296 |
| | Avg. | **0.157** | 0.243 | 0.900 | 0.762 | 0.304 | 0.362 | 0.449 | 0.490 | - | - | 0.161 | **0.243** | 0.166 | 0.262 | - | - | - | - | 0.163 | 0.259 | 0.168 | 0.263 | 0.198 | 0.301 | 0.167 | 0.262 |
| Weather | 96 | 0.147 | 0.187 | 0.260 | 0.309 | 0.181 | 0.232 | 0.201 | 0.247 | **0.146** | 0.182 | 0.149 | **0.179** | 0.157 | 0.208 | 0.159 | 0.213 | 0.150 | 0.183 | 0.155 | 0.212 | 0.150 | 0.197 | 0.174 | 0.222 | 0.149 | 0.205 |
| | 192 | **0.190** | 0.229 | 0.298 | 0.336 | 0.284 | 0.326 | 0.260 | 0.294 | 0.192 | 0.228 | 0.192 | **0.223** | 0.207 | 0.256 | 0.215 | 0.266 | 0.197 | 0.230 | 0.196 | 0.245 | 0.197 | 0.241 | 0.229 | 0.265 | 0.195 | 0.248 |
| | 336 | **0.233** | 0.263 | 0.347 | 0.364 | 0.369 | 0.376 | 0.314 | 0.330 | 0.248 | 0.270 | 0.245 | 0.265 | 0.259 | 0.295 | 0.291 | 0.322 | 0.255 | 0.272 | 0.252 | 0.289 | 0.248 | 0.281 | 0.282 | 0.304 | 0.254 | 0.293 |
| | 720 | **0.297** | **0.308** | 0.400 | 0.401 | 0.469 | 0.432 | 0.390 | 0.392 | 0.338 | 0.330 | 0.310 | 0.312 | 0.327 | 0.342 | 0.415 | 0.400 | 0.344 | 0.329 | 0.323 | 0.334 | 0.326 | 0.334 | 0.360 | 0.352 | 0.346 | 0.354 |
| | Avg. | **0.217** | 0.247 | 0.326 | 0.353 | 0.326 | 0.342 | 0.291 | 0.306 | 0.231 | 0.253 | 0.224 | **0.245** | 0.238 | 0.275 | 0.256 | 0.288 | 0.237 | 0.254 | 0.232 | 0.270 | 0.230 | 0.263 | 0.261 | 0.286 | 0.236 | 0.275 |
| # Wins | | **19** | **14** | 0 | 0 | 0 | 0 | 0 | 0 | 3 | 1 | 0 | 6 | 0 | 0 | 0 | 0 | 1 | 5 | 1 | 0 | 0 | 0 | 0 | 0 | 0 | 0 |

with the Seasonal Naive MAE being defined as

$$\text{MAE}_{\text{seasonal naive}} = \frac{1}{T-m} \sum_{t=m+1}^{T} |y_t - y_{t-m}| \qquad (17)$$

where $T$ is the length of the *training* split of the series, $y_t$ is the value of the series at time $t$, and $m$ is the seasonal period.

**CRPS** (Gneiting & Raftery, 2007) is a scoring rule for probabilistic forecasting. Given a predicted distribution with CDF $F$ and a ground truth value $y$, the CRPS is defined as:

$$\text{CRPS}(F, y) = \int_0^1 2\Lambda_\alpha(F^{-1}(\alpha), y) \, d\alpha, \qquad (18)$$

where $\Lambda_\alpha(q, y)$ denotes the quantile loss.

In practice, CRPS is approximated by a discrete sum over a finite set of quantile levels. This approximation, often referred to as the mean weighted quantile loss (Park et al., 2021), is given by:

$$\text{CRPS} \approx \frac{1}{K} \sum_{k=1}^{K} \text{wQL}[\alpha_k], \tag{19}$$

where $K$ is the number of quantile levels, and $\{\alpha_1, \alpha_2, \ldots, \alpha_K\}$ are the selected quantile levels. The weighted quantile loss wQL$[\alpha]$ for each quantile level $\alpha$ is calculated as:

$$\text{wQL}[\alpha] = 2 \frac{\sum_t \Lambda_\alpha(\hat{q}_t(\alpha), y_t)}{\sum_t |y_t|}, \tag{20}$$

where $\hat{q}_t(\alpha)$ is the predicted $\alpha$-quantile at time step $t$.

As in GIFT-Eval, both MASE and CRPS are further normalized by the performance of the Seasonal Naive forecast on the *test* split.

**Full Results**   Following prior work (Auer et al., 2025; Liu et al., 2025b), we report the aggregated results in the main text, while the full results are provided in the project repository.

### D.3. Imputation

**Benchmarks**   Following prior work (Liu et al., 2024b; Zhang et al., 2025), we evaluate ZEUS on the same six datasets used for point forecasting. We mask $\{12.5\%, 25\%, 37.5\%, 50\%\}$ time steps in sequences of length 192.

Existing TSFMs are typically evaluated under patch-wise missing patterns. For example, MOMENT uses a patch size of 8 and therefore only considers missing segments of length 8 (Goswami et al., 2024), while Timer uses a patch size of 24 and considers missing segments of length 24 (Liu et al., 2024b). However, real-world time series exhibit more diverse missing behaviors, including both point-wise missing and contiguous segment missing with variable lengths (Yu et al., 2025b). To better reflect practical scenarios, we adopt two masking strategies:

1. **Random masking** simulates point-wise missing observations that occur sporadically in real-world time series. Such missing values often arise from packet loss during data transmission or occasional logging failures. This setting has been widely adopted in prior work (Wu et al., 2023; donghao & xue, 2024; Fu et al., 2026).

2. **Block masking**, where the lengths of contiguous missing segments are sampled from a geometric distribution with $p = 0.125$. This strategy reflects structured missing patterns in practice (e.g., sensor outages or system downtimes). The heavy-tailed nature of the geometric distribution aligns well with real-world missing behaviors, where short gaps are common but extended missing segments can still arise occasionally. We deliberately choose a small $p$ to create more challenging missing patterns and to clearly distinguish this setting from random masking. The probability distribution of missing segment lengths is shown in Figure 9.

**Baselines**   The baselines for the imputation task include TSFMs, namely **MOMENT** (Goswami et al., 2024), **Timer** (Liu et al., 2024b) and **UniTS** (Gao et al., 2024), as well as advanced task-specific models, including **GPT4TS** (Zhou et al., 2023), **ModernTCN** (donghao & xue, 2024), **TimesNet** (Wu et al., 2023), **PatchTST** (Nie et al., 2023), and **DLinear** (Zeng et al., 2023).

**Metrics**   Consistent with the point forecasting setting, we adopt **Mean Squared Error (MSE)** and **Mean Absolute Error (MAE)** as evaluation metrics.

**Full Results**   The full results of imputation evaluation are provided in Table D.3.

*Table 8.* Imputation results under `random` and `block` masking, where {12.5%, 25%, 37.5%, 50%} of time steps are masked.

| Models | | | ZEUS (Ours) | | MOMENT (2024) | | Timer (2024b) | | UniTS (2024) | | GPT4TS (2023) | | ModernTCN (2024) | | TimesNet (2023) | | PatchTST (2023) | | DLinear (2023) | |
|---|---|---|---|---|---|---|---|---|---|---|---|---|---|---|---|---|---|---|---|---|
| | | Metrics | MSE | MAE | MSE | MAE | MSE | MAE | MSE | MAE | MSE | MAE | MSE | MAE | MSE | MAE | MSE | MAE | MSE | MAE |
| ETTh1 | Random | 12.5% | 0.064 | 0.159 | 0.300 | 0.352 | 0.429 | 0.423 | 0.672 | 0.548 | 0.071 | 0.183 | 0.067 | 0.178 | 0.059 | 0.163 | 0.099 | 0.209 | 0.111 | 0.233 |
| | | 25% | 0.071 | 0.167 | 0.351 | 0.382 | 0.464 | 0.441 | 0.761 | 0.602 | 0.089 | 0.201 | 0.077 | 0.193 | 0.078 | 0.187 | 0.119 | 0.230 | 0.147 | 0.267 |
| | | 37.5% | 0.082 | 0.179 | 0.406 | 0.412 | 0.502 | 0.460 | 0.861 | 0.635 | 0.110 | 0.223 | 0.091 | 0.209 | 0.098 | 0.208 | 0.140 | 0.248 | 0.181 | 0.295 |
| | | 50% | 0.099 | 0.195 | 0.472 | 0.445 | 0.541 | 0.478 | 0.856 | 0.671 | 0.141 | 0.253 | 0.110 | 0.226 | 0.119 | 0.232 | 0.165 | 0.268 | 0.217 | 0.322 |
| | | Avg. | 0.079 | 0.175 | 0.382 | 0.398 | 0.484 | 0.451 | 0.788 | 0.614 | 0.103 | 0.215 | 0.086 | 0.202 | 0.089 | 0.198 | 0.131 | 0.239 | 0.167 | 0.279 |
| | Block | 12.5% | 0.102 | 0.188 | 0.329 | 0.370 | 0.459 | 0.436 | 0.628 | 0.532 | 0.095 | 0.211 | 0.083 | 0.201 | 0.078 | 0.187 | 0.179 | 0.271 | 0.215 | 0.316 |
| | | 25% | 0.110 | 0.196 | 0.377 | 0.397 | 0.488 | 0.451 | 0.765 | 0.605 | 0.122 | 0.234 | 0.100 | 0.219 | 0.098 | 0.208 | 0.185 | 0.276 | 0.229 | 0.329 |
| | | 37.5% | 0.117 | 0.204 | 0.437 | 0.428 | 0.525 | 0.469 | 0.811 | 0.639 | 0.141 | 0.250 | 0.105 | 0.224 | 0.124 | 0.228 | 0.198 | 0.286 | 0.249 | 0.343 |
| | | 50% | 0.132 | 0.218 | 0.504 | 0.461 | 0.557 | 0.485 | 0.863 | 0.674 | 0.182 | 0.276 | 0.129 | 0.245 | 0.142 | 0.247 | 0.209 | 0.297 | 0.271 | 0.359 |
| | | Avg. | 0.115 | 0.202 | 0.412 | 0.414 | 0.507 | 0.460 | 0.767 | 0.613 | 0.135 | 0.243 | 0.104 | 0.222 | 0.111 | 0.218 | 0.193 | 0.283 | 0.241 | 0.337 |
| ETTh2 | Random | 12.5% | 0.051 | 0.129 | 0.129 | 0.241 | 0.172 | 0.271 | 0.289 | 0.381 | 0.050 | 0.149 | 0.048 | 0.148 | 0.045 | 0.140 | 0.059 | 0.156 | 0.104 | 0.219 |
| | | 25% | 0.053 | 0.133 | 0.155 | 0.268 | 0.177 | 0.279 | 0.438 | 0.474 | 0.060 | 0.164 | 0.054 | 0.157 | 0.054 | 0.154 | 0.065 | 0.165 | 0.132 | 0.249 |
| | | 37.5% | 0.057 | 0.137 | 0.179 | 0.289 | 0.184 | 0.287 | 0.659 | 0.583 | 0.068 | 0.177 | 0.059 | 0.165 | 0.063 | 0.167 | 0.071 | 0.173 | 0.162 | 0.276 |
| | | 50% | 0.062 | 0.144 | 0.202 | 0.307 | 0.193 | 0.296 | 0.971 | 0.710 | 0.081 | 0.193 | 0.070 | 0.178 | 0.074 | 0.183 | 0.079 | 0.183 | 0.189 | 0.300 |
| | | Avg. | 0.056 | 0.136 | 0.166 | 0.276 | 0.182 | 0.283 | 0.589 | 0.537 | 0.065 | 0.171 | 0.058 | 0.162 | 0.059 | 0.161 | 0.069 | 0.169 | 0.147 | 0.261 |
| | Block | 12.5% | 0.063 | 0.145 | 0.169 | 0.268 | 0.184 | 0.280 | 0.326 | 0.401 | 0.070 | 0.179 | 0.067 | 0.177 | 0.067 | 0.172 | 0.088 | 0.193 | 0.244 | 0.330 |
| | | 25% | 0.064 | 0.147 | 0.183 | 0.286 | 0.187 | 0.286 | 0.470 | 0.488 | 0.076 | 0.188 | 0.072 | 0.184 | 0.074 | 0.181 | 0.090 | 0.194 | 0.275 | 0.348 |
| | | 37.5% | 0.068 | 0.152 | 0.199 | 0.302 | 0.194 | 0.293 | 0.683 | 0.591 | 0.085 | 0.199 | 0.074 | 0.186 | 0.080 | 0.189 | 0.094 | 0.199 | 0.288 | 0.361 |
| | | 50% | 0.073 | 0.159 | 0.218 | 0.319 | 0.201 | 0.300 | 0.991 | 0.715 | 0.095 | 0.210 | 0.079 | 0.192 | 0.089 | 0.198 | 0.099 | 0.206 | 0.304 | 0.374 |
| | | Avg. | 0.067 | 0.151 | 0.192 | 0.294 | 0.192 | 0.290 | 0.618 | 0.549 | 0.082 | 0.194 | 0.073 | 0.185 | 0.078 | 0.185 | 0.093 | 0.198 | 0.278 | 0.353 |
| ETTm1 | Random | 12.5% | 0.034 | 0.110 | 0.185 | 0.275 | 0.755 | 0.523 | 0.617 | 0.513 | 0.046 | 0.144 | 0.041 | 0.136 | 0.037 | 0.127 | 0.046 | 0.140 | 0.056 | 0.163 |
| | | 25% | 0.036 | 0.113 | 0.242 | 0.316 | 0.688 | 0.508 | 0.717 | 0.579 | 0.060 | 0.165 | 0.047 | 0.146 | 0.047 | 0.143 | 0.056 | 0.154 | 0.074 | 0.189 |
| | | 37.5% | 0.039 | 0.117 | 0.303 | 0.355 | 0.644 | 0.499 | 0.763 | 0.615 | 0.069 | 0.175 | 0.052 | 0.153 | 0.056 | 0.157 | 0.063 | 0.164 | 0.094 | 0.213 |
| | | 50% | 0.044 | 0.126 | 0.371 | 0.392 | 0.616 | 0.494 | 0.821 | 0.653 | 0.083 | 0.190 | 0.065 | 0.173 | 0.073 | 0.179 | 0.068 | 0.171 | 0.120 | 0.239 |
| | | Avg. | 0.038 | 0.116 | 0.275 | 0.335 | 0.676 | 0.506 | 0.730 | 0.590 | 0.065 | 0.169 | 0.051 | 0.152 | 0.053 | 0.152 | 0.058 | 0.157 | 0.086 | 0.201 |
| | Block | 12.5% | 0.056 | 0.134 | 0.244 | 0.310 | 0.772 | 0.531 | 0.607 | 0.509 | 0.076 | 0.180 | 0.058 | 0.164 | 0.061 | 0.158 | 0.113 | 0.208 | 0.163 | 0.273 |
| | | 25% | 0.061 | 0.138 | 0.292 | 0.343 | 0.710 | 0.517 | 0.722 | 0.582 | 0.086 | 0.193 | 0.069 | 0.175 | 0.075 | 0.176 | 0.116 | 0.212 | 0.177 | 0.282 |
| | | 37.5% | 0.065 | 0.143 | 0.347 | 0.376 | 0.669 | 0.508 | 0.777 | 0.621 | 0.100 | 0.205 | 0.078 | 0.185 | 0.082 | 0.182 | 0.117 | 0.211 | 0.192 | 0.295 |
| | | 50% | 0.074 | 0.152 | 0.406 | 0.409 | 0.640 | 0.502 | 0.831 | 0.657 | 0.113 | 0.218 | 0.091 | 0.200 | 0.087 | 0.194 | 0.121 | 0.215 | 0.211 | 0.311 |
| | | Avg. | 0.064 | 0.142 | 0.322 | 0.360 | 0.698 | 0.515 | 0.734 | 0.592 | 0.094 | 0.199 | 0.074 | 0.181 | 0.076 | 0.178 | 0.117 | 0.212 | 0.186 | 0.29 |
| ETTm2 | Random | 12.5% | 0.024 | 0.080 | 0.076 | 0.180 | 0.126 | 0.239 | 0.203 | 0.324 | 0.027 | 0.104 | 0.026 | 0.101 | 0.025 | 0.100 | 0.029 | 0.104 | 0.064 | 0.169 |
| | | 25% | 0.025 | 0.081 | 0.095 | 0.206 | 0.123 | 0.237 | 0.348 | 0.428 | 0.031 | 0.115 | 0.030 | 0.110 | 0.031 | 0.112 | 0.032 | 0.110 | 0.099 | 0.211 |
| | | 37.5% | 0.028 | 0.085 | 0.112 | 0.227 | 0.124 | 0.239 | 0.575 | 0.551 | 0.036 | 0.124 | 0.034 | 0.118 | 0.033 | 0.118 | 0.037 | 0.119 | 0.115 | 0.230 |
| | | 50% | 0.031 | 0.091 | 0.128 | 0.244 | 0.127 | 0.242 | 0.893 | 0.687 | 0.042 | 0.134 | 0.039 | 0.126 | 0.038 | 0.127 | 0.04 | 0.126 | 0.128 | 0.244 |
| | | Avg. | 0.027 | 0.084 | 0.103 | 0.214 | 0.125 | 0.239 | 0.505 | 0.498 | 0.034 | 0.119 | 0.032 | 0.114 | 0.032 | 0.114 | 0.035 | 0.115 | 0.102 | 0.214 |
| | Block | 12.5% | 0.032 | 0.095 | 0.115 | 0.213 | 0.133 | 0.245 | 0.249 | 0.353 | 0.044 | 0.139 | 0.043 | 0.136 | 0.040 | 0.131 | 0.049 | 0.138 | 0.170 | 0.268 |
| | | 25% | 0.033 | 0.097 | 0.117 | 0.225 | 0.130 | 0.243 | 0.382 | 0.446 | 0.045 | 0.141 | 0.044 | 0.138 | 0.043 | 0.134 | 0.050 | 0.140 | 0.183 | 0.286 |
| | | 37.5% | 0.035 | 0.100 | 0.127 | 0.240 | 0.130 | 0.244 | 0.602 | 0.561 | 0.048 | 0.147 | 0.045 | 0.140 | 0.046 | 0.140 | 0.053 | 0.144 | 0.198 | 0.300 |
| | | 50% | 0.038 | 0.105 | 0.140 | 0.255 | 0.132 | 0.246 | 0.913 | 0.692 | 0.053 | 0.153 | 0.052 | 0.149 | 0.050 | 0.146 | 0.056 | 0.149 | 0.212 | 0.312 |
| | | Avg. | 0.035 | 0.099 | 0.125 | 0.233 | 0.131 | 0.245 | 0.537 | 0.513 | 0.048 | 0.145 | 0.046 | 0.141 | 0.045 | 0.138 | 0.052 | 0.143 | 0.191 | 0.292 |
| ECL | Random | 12.5% | 0.037 | 0.120 | 0.206 | 0.335 | 0.309 | 0.413 | 0.729 | 0.689 | 0.104 | 0.225 | 0.071 | 0.188 | 0.097 | 0.217 | 0.068 | 0.183 | 0.076 | 0.193 |
| | | 25% | 0.041 | 0.126 | 0.262 | 0.388 | 0.369 | 0.467 | 0.926 | 0.790 | 0.111 | 0.233 | 0.093 | 0.219 | 0.102 | 0.221 | 0.080 | 0.200 | 0.100 | 0.226 |
| | | 37.5% | 0.047 | 0.135 | 0.330 | 0.444 | 0.443 | 0.528 | 0.935 | 0.797 | 0.117 | 0.238 | 0.114 | 0.244 | 0.106 | 0.225 | 0.099 | 0.224 | 0.122 | 0.252 |
| | | 50% | 0.056 | 0.148 | 0.417 | 0.510 | 0.527 | 0.588 | 0.960 | 0.806 | 0.123 | 0.245 | 0.138 | 0.269 | 0.110 | 0.229 | 0.113 | 0.240 | 0.147 | 0.278 |
| | | Avg. | 0.045 | 0.132 | 0.304 | 0.419 | 0.412 | 0.499 | 0.888 | 0.771 | 0.114 | 0.235 | 0.104 | 0.230 | 0.104 | 0.223 | 0.090 | 0.212 | 0.111 | 0.237 |
| | Block | 12.5% | 0.050 | 0.135 | 0.226 | 0.347 | 0.345 | 0.438 | 0.617 | 0.620 | 0.114 | 0.231 | 0.107 | 0.228 | 0.105 | 0.223 | 0.100 | 0.217 | 0.124 | 0.244 |
| | | 25% | 0.054 | 0.140 | 0.279 | 0.396 | 0.399 | 0.485 | 0.917 | 0.785 | 0.118 | 0.240 | 0.114 | 0.234 | 0.108 | 0.227 | 0.112 | 0.233 | 0.140 | 0.264 |
| | | 37.5% | 0.059 | 0.148 | 0.352 | 0.456 | 0.469 | 0.541 | 0.947 | 0.800 | 0.128 | 0.257 | 0.126 | 0.251 | 0.111 | 0.230 | 0.123 | 0.247 | 0.158 | 0.283 |
| | | 50% | 0.067 | 0.159 | 0.451 | 0.529 | 0.551 | 0.599 | 0.964 | 0.808 | 0.136 | 0.268 | 0.141 | 0.276 | 0.115 | 0.236 | 0.130 | 0.254 | 0.179 | 0.304 |
| | | Avg. | 0.058 | 0.146 | 0.327 | 0.432 | 0.441 | 0.516 | 0.861 | 0.753 | 0.124 | 0.249 | 0.122 | 0.247 | 0.110 | 0.229 | 0.116 | 0.238 | 0.150 | 0.274 |
| Weather | Random | 12.5% | 0.027 | 0.032 | 0.067 | 0.112 | 0.111 | 0.162 | 0.135 | 0.206 | 0.030 | 0.062 | 0.030 | 0.056 | 0.029 | 0.058 | 0.031 | 0.057 | 0.064 | 0.169 |
| | | 25% | 0.029 | 0.033 | 0.077 | 0.134 | 0.11 | 0.164 | 0.182 | 0.265 | 0.034 | 0.069 | 0.032 | 0.061 | 0.032 | 0.064 | 0.034 | 0.061 | 0.099 | 0.211 |
| | | 37.5% | 0.031 | 0.035 | 0.088 | 0.152 | 0.111 | 0.169 | 0.226 | 0.313 | 0.037 | 0.074 | 0.035 | 0.066 | 0.036 | 0.070 | 0.037 | 0.066 | 0.115 | 0.230 |
| | | 50% | 0.034 | 0.038 | 0.099 | 0.169 | 0.115 | 0.176 | 0.285 | 0.366 | 0.043 | 0.081 | 0.038 | 0.071 | 0.039 | 0.076 | 0.041 | 0.072 | 0.128 | 0.244 |
| | | Avg. | 0.030 | 0.035 | 0.083 | 0.142 | 0.112 | 0.168 | 0.207 | 0.288 | 0.036 | 0.072 | 0.034 | 0.064 | 0.034 | 0.067 | 0.036 | 0.064 | 0.102 | 0.214 |
| | Block | 12.5% | 0.033 | 0.040 | 0.083 | 0.133 | 0.119 | 0.171 | 0.136 | 0.209 | 0.039 | 0.075 | 0.042 | 0.079 | 0.037 | 0.074 | 0.050 | 0.088 | 0.087 | 0.158 |
| | | 25% | 0.035 | 0.041 | 0.088 | 0.147 | 0.118 | 0.172 | 0.184 | 0.268 | 0.042 | 0.080 | 0.045 | 0.084 | 0.042 | 0.080 | 0.048 | 0.081 | 0.091 | 0.164 |
| | | 37.5% | 0.036 | 0.043 | 0.096 | 0.161 | 0.118 | 0.176 | 0.233 | 0.318 | 0.046 | 0.087 | 0.046 | 0.085 | 0.044 | 0.084 | 0.050 | 0.083 | 0.095 | 0.172 |
| | | 50% | 0.039 | 0.045 | 0.106 | 0.176 | 0.120 | 0.181 | 0.289 | 0.368 | 0.050 | 0.092 | 0.049 | 0.091 | 0.048 | 0.090 | 0.054 | 0.089 | 0.100 | 0.180 |
| | | Avg. | 0.036 | 0.042 | 0.093 | 0.154 | 0.119 | 0.175 | 0.211 | 0.291 | 0.044 | 0.084 | 0.046 | 0.085 | 0.043 | 0.082 | 0.051 | 0.085 | 0.093 | 0.169 |

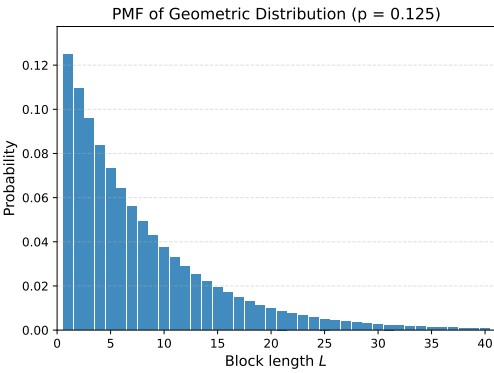

*Figure 9.* PMF of the geometric distribution used to sample missing segment lengths. The expected block length is 8, and with 99% probability the block length is smaller than 35.

## D.4. Anomaly Detection

**Benchmarks**  We evaluate the anomaly detection task on the UCR Anomaly Archive (Wu & Keogh, 2021), which consists of 250 tasks spanning diverse domains such as medicine, sports, entomology, and space science. The dataset contains both realistic and synthetic anomalies, addressing several limitations of earlier anomaly detection benchmarks. We conduct experiments on a subset of 42 tasks selected by MOMENT[2], which covers a wide range of domains and exhibits sufficient data diversity (Goswami et al., 2024).

Previous work typically evaluates models using a fixed window size. However, datasets in the UCR archive exhibit substantial variation in periodicity, making a fixed window potentially detrimental to model performance. Therefore, we treat the window size as a hyperparameter and select the optimal value via search from $\{64, 256, 512, 1024, 2048\}$.

**Baselines**  The baselines for anomaly detection include *TSFMs*—**MOMENT** (Goswami et al., 2024), **Timer** (Liu et al., 2024b), and **UniTS** (Gao et al., 2024)—as well as *task-specific models*, including **ModernTCN** (donghao & xue, 2024), **GPT4TS** (Zhou et al., 2023), **TimesNet** (Wu et al., 2023), **PatchTST** (Nie et al., 2023), and **Anomaly Transformer** (Xu et al., 2022).

**Metrics**  Following prior work (Wu et al., 2023; Goswami et al., 2024), we use the adjusted F1 score as the evaluation metric. Specifically, point adjustment is applied such that if an anomaly is detected at any time point within a ground-truth anomalous segment, the entire segment is considered correctly detected. Precision and recall are then computed based on these adjusted predictions, and the adjusted F1 score is calculated accordingly. Compared to the standard F1 score, the adjusted F1 score better reflects detection quality at the event level and reduces over-penalization caused by slight temporal misalignment.

**Full Results**  Table 9 presents the full results of anomaly detection.

## D.5. Classification

**Benchmarks**  We conduct evaluations on the UEA classification archive (Bagnall et al., 2018), which comprises 30 multi-variate time series classification datasets. Since four datasets (namely `CharacterTrajectories`, `InsectWingBeat`, `JapaneseVowels`, and `SpokenArabicDigits`) contain sequences with unequal lengths, some baselines are incompatible with them and are marked as NA in the corresponding works. To enable a fair comparison using dataset-averaged metrics, we conduct evaluations only on the 26 datasets with equal-length sequences.

While prior work on classification tasks relies on fine-tuning (Gao et al., 2024; Zhang et al., 2025) or linear probing (Goswami et al., 2024), ZEUS is evaluated in a tuning-free setting. Accordingly, we use a non-parametric 1-nearest neighbor

---

[2]Although MOMENT originally states that 45 tasks are selected, only 42 tasks are reported in its published results. We therefore follow the reported tables and conduct our evaluation on these 42 tasks.

*Table 9.* Adjusted best F1 score on 42 datasets sampled from the UCR Anomaly archive.

| Models | TSFMs (Zero-shot) | | | | Task-Specific Models (Supervised) | | | | |
|---|---|---|---|---|---|---|---|---|---|
| | ZEUS (Ours) | MOMENT (2024) | Timer (2024b) | UniTS (2024) | ModernTCN (2024) | GPT4TS (2023) | TimesNet (2023) | PatchTST (2023) | Ano. Trans. (2022) |
| 1sddb40 | 0.966 | 0.680 | 0.578 | 0.915 | **1.000** | 0.930 | 0.972 | 0.777 | 0.858 |
| BIDMC1 | **1.000** | **1.000** | **1.000** | 0.858 | 0.998 | **1.000** | **1.000** | **1.000** | 0.990 |
| CHARISfive | 0.545 | 0.375 | 0.007 | 0.008 | 0.185 | 0.108 | 0.937 | **1.000** | 0.968 |
| CHARISten | **0.882** | 0.504 | 0.310 | 0.667 | 0.800 | 0.352 | 0.382 | 0.795 | 0.144 |
| CIMIS44AirTemperature3 | **1.000** | 0.350 | **1.000** | 0.906 | 0.361 | 0.180 | 0.906 | 0.906 | 0.085 |
| CIMIS44AirTemperature5 | **1.000** | 0.615 | 0.889 | **1.000** | **1.000** | 0.200 | 0.970 | 0.897 | 0.390 |
| ECG2 | **1.000** | **1.000** | **1.000** | 0.957 | 0.990 | 0.900 | **1.000** | **1.000** | **1.000** |
| ECG3 | 0.962 | 0.823 | 0.195 | 0.562 | 0.823 | 0.840 | 0.990 | **0.995** | 0.360 |
| Fantasia | 0.984 | 0.943 | 0.989 | 0.968 | **0.997** | 0.870 | 0.990 | 0.992 | 0.971 |
| GP711MarkerLFM5z4 | **1.000** | 0.871 | 0.535 | 0.843 | 0.959 | 0.819 | 0.987 | **1.000** | 0.930 |
| GP711MarkerLFM5z5 | **1.000** | **1.000** | **1.000** | 0.963 | 0.889 | 0.929 | 0.950 | **1.000** | 0.852 |
| InternalBleeding4 | **1.000** | **1.000** | **1.000** | **1.000** | 0.993 | 0.992 | **1.000** | **1.000** | 0.992 |
| InternalBleeding5 | **1.000** | **1.000** | **1.000** | **1.000** | **1.000** | 0.994 | **1.000** | **1.000** | 0.940 |
| Italianpowerdemand | 0.857 | 0.525 | 0.080 | **0.990** | 0.445 | 0.141 | 0.353 | 0.630 | 0.010 |
| Lab2Cmac011215EPG5 | 0.956 | 0.858 | 0.389 | 0.498 | **0.996** | 0.847 | 0.990 | **0.996** | 0.990 |
| Lab2Cmac011215EPG6 | 0.488 | 0.020 | 0.071 | 0.127 | 0.340 | 0.100 | 0.151 | 0.381 | **0.806** |
| MesoplodonDensirostris | **1.000** | 0.967 | **1.000** | **1.000** | **1.000** | **1.000** | **1.000** | 0.994 | **1.000** |
| PowerDemand1 | 0.953 | 0.809 | 0.956 | 0.955 | 0.887 | 0.760 | 0.991 | **0.997** | 0.870 |
| TkeepFirstMARS | 0.072 | 0.046 | 0.021 | 0.019 | **0.375** | 0.020 | 0.082 | 0.226 | 0.175 |
| TkeepSecondMARS | **1.000** | **1.000** | 0.153 | **1.000** | 0.645 | 0.417 | 0.741 | **1.000** | 0.830 |
| WalkingAceleration5 | **1.000** | **1.000** | 0.808 | 0.175 | 0.959 | 0.870 | 0.930 | **1.000** | 0.990 |
| apneaecg | 0.802 | 0.951 | 0.267 | 0.997 | 0.958 | 0.919 | 0.978 | **1.000** | 0.400 |
| apneaecg2 | 0.952 | 0.990 | **1.000** | 0.997 | 0.990 | **1.000** | 0.980 | **1.000** | 0.817 |
| gait1 | 0.940 | 0.793 | 0.418 | 0.714 | 0.971 | 0.739 | 0.715 | 0.955 | **1.000** |
| gaitHunt1 | 0.849 | 0.500 | 0.045 | 0.721 | 0.917 | 0.579 | 0.840 | **0.969** | 0.080 |
| insectEPG2 | **1.000** | 0.124 | 0.243 | **1.000** | 0.633 | 0.810 | **1.000** | 0.943 | 0.420 |
| insectEPG4 | 0.962 | 0.990 | 0.187 | 0.123 | 0.610 | 0.210 | **1.000** | **1.000** | 0.980 |
| ltstdbs30791AS | **1.000** | **1.000** | **1.000** | **1.000** | **1.000** | **1.000** | **1.000** | **1.000** | **1.000** |
| mit14046longtermecg | 0.951 | 0.406 | 0.843 | 0.950 | 0.962 | 0.979 | **1.000** | 0.993 | 0.858 |
| park3m | **0.999** | 0.723 | 0.973 | **0.999** | 0.945 | 0.987 | 0.943 | 0.943 | 0.987 |
| qtdbSel1005V | 0.878 | 0.811 | 0.837 | 0.945 | 0.763 | **0.986** | 0.871 | 0.932 | 0.457 |
| qtdbSel100MLII | 0.891 | 0.883 | 0.888 | 0.988 | 0.970 | **0.996** | 0.929 | 0.958 | 0.809 |
| resperation1 | **0.930** | 0.040 | 0.051 | 0.003 | 0.014 | 0.052 | 0.920 | 0.219 | 0.003 |
| s20101mML2 | **1.000** | 0.687 | 0.995 | 0.995 | **1.000** | 0.996 | **1.000** | **1.000** | **1.000** |
| sddb49 | 0.998 | **1.000** | **1.000** | **1.000** | **1.000** | 0.940 | **1.000** | **1.000** | 0.890 |
| sel840mECG1 | **1.000** | 0.991 | 0.980 | **1.000** | 0.971 | 0.999 | 0.960 | 0.993 | 0.894 |
| sel840mECG2 | **0.993** | 0.937 | 0.754 | 0.842 | 0.745 | 0.983 | **0.993** | 0.951 | 0.597 |
| tilt12744mtable | 0.244 | 0.206 | 0.079 | 0.159 | 0.128 | 0.612 | **0.734** | 0.003 | 0.070 |
| tilt12754table | **0.952** | 0.754 | 0.593 | 0.906 | 0.490 | 0.877 | 0.445 | 0.881 | 0.715 |
| tiltAPB2 | 0.990 | 0.829 | 0.657 | 0.943 | 0.989 | 0.990 | 0.893 | **0.999** | 0.920 |
| tiltAPB3 | **0.955** | 0.080 | 0.095 | 0.198 | 0.464 | 0.285 | 0.819 | 0.921 | 0.170 |
| weallwalk | 0.857 | **0.984** | 0.231 | 0.349 | **0.984** | 0.199 | 0.612 | 0.600 | 0.179 |
| Avg. F1 Score | **0.900** | 0.716 | 0.598 | 0.744 | 0.789 | 0.676 | 0.856 | 0.877 | 0.651 |
| # Wins | **21** | 10 | 10 | 11 | 12 | 6 | 13 | 20 | 6 |

(1-NN) classifier for evaluation, with optional PCA whitening applied as a feature normalization step.

**Baselines** The baselines include two *TSFMs*, **MOMENT** (Goswami et al., 2024) and **UniTS** (Gao et al., 2024), a *pretrained classification model* **VQ-shape** (Wen et al., 2024), and *task-specific models* including **ModernTCN** (donghao & xue, 2024), **SVP-T** (Zuo et al., 2023), **GPT4TS** (Zhou et al., 2023), **TimesNet** (Wu et al., 2023), **PatchTST** (Nie et al., 2023); and *classical methods* including **Rocket** (Dempster et al., 2020), **Shapelet Transformation with Random Forest (STRF)** (Hills et al., 2014), and **DTW** (Chen et al., 2013).

All pretrained baselines are fine-tuned in the evaluation of their original works and do not support a tuning-free setting. Specifically, MOMENT performs linear probing using an SVM classifier on pooled representations, VQ-Shape trains a linear head for downstream classification, and UniTS fine-tunes prompt tokens for task adaptation. To compare the tuning-free performance with other pretrained baselines, we replace the SVM classifier in MOMENT with an 1-NN classifier and re-evaluate the results, denoted as MOMENT[†].

**Metrics** Following prior work (Goswami et al., 2024), we use the accuracy as the evaluation metric.

**Full Results** The full results for classification tasks are provided in Table 10.

*Table 10.* Classification accuracy of methods across 26 UEA datasets. † denotes linear probing, and ‡ denotes prompt-based fine-tuning.

| | TSFMs (1-NN) | | Pretrained Models (Fine-tuned) | | | | Task-Specific Models (Supervised) | | | | | Classical | | |
|---|---|---|---|---|---|---|---|---|---|---|---|---|---|---|
| | **ZEUS (Ours)** | **MOMENT** (2024) | **ZEUS**[†] **(Ours)** | **MOMENT**[†] (2024) | **UniTS**[‡] (2024) | **VQ-Shape**[†] (2024) | **ModernTCN** (2024) | **SVP-T** (2023) | **GPT4TS** (2023) | **TimesNet** (2023) | **PatchTST** (2023) | **Rocket** (2020) | **STRF** (2014) | **DTW** (2013) |
| ArticularyWordRecognition | 0.977 | 0.987 | 0.990 | 0.990 | 0.927 | 0.987 | 0.983 | 0.993 | 0.933 | 0.973 | 0.927 | 0.996 | 0.917 | 0.987 |
| AtrialFibrillation | 0.467 | 0.067 | 0.533 | 0.200 | 0.133 | 0.520 | 0.467 | 0.400 | 0.333 | 0.333 | 0.333 | 0.249 | 0.267 | 0.200 |
| BasicMotions | 0.975 | 1.000 | 0.975 | 1.000 | 0.600 | 0.910 | 0.975 | 1.000 | 0.925 | 0.975 | 0.700 | 0.990 | 0.925 | 0.975 |
| Cricket | 0.986 | 0.903 | 1.000 | 0.986 | 0.958 | 0.978 | 0.958 | 1.000 | 0.847 | 0.903 | 0.889 | 1.000 | 0.944 | 1.000 |
| DuckDuckGeese | 0.500 | 0.400 | 0.560 | 0.600 | 0.320 | 0.360 | 0.560 | 0.700 | 0.560 | 0.580 | 0.220 | 0.461 | 0.380 | 0.600 |
| EigenWorms | 0.954 | 0.626 | 0.970 | 0.809 | 0.710 | 0.603 | 0.672 | 0.923 | 0.542 | 0.550 | 0.415 | 0.863 | 0.672 | 0.618 |
| Epilepsy | 0.986 | 0.971 | 1.000 | 0.993 | 0.942 | 0.893 | 0.957 | 0.986 | 0.855 | 0.877 | 0.913 | 0.991 | 0.978 | 0.964 |
| ERing | 0.926 | 0.944 | 0.959 | 0.963 | 0.830 | 0.960 | 0.952 | 0.937 | 0.948 | 0.927 | 0.937 | 0.981 | 0.889 | 0.133 |
| EthanolConcentration | 0.289 | 0.171 | 0.395 | 0.357 | 0.259 | 0.325 | 0.363 | 0.331 | 0.255 | 0.285 | 0.259 | 0.447 | 0.677 | 0.323 |
| FaceDetection | 0.552 | 0.509 | 0.639 | 0.633 | 0.549 | 0.653 | 0.708 | 0.512 | 0.656 | 0.677 | 0.668 | 0.694 | 0.567 | 0.529 |
| FingerMovements | 0.600 | 0.490 | 0.650 | 0.490 | 0.520 | 0.640 | 0.670 | 0.600 | 0.570 | 0.530 | 0.580 | 0.553 | 0.500 | 0.530 |
| HandMovementDirection | 0.365 | 0.216 | 0.419 | 0.324 | 0.365 | 0.546 | 0.527 | 0.392 | 0.473 | 0.595 | 0.514 | 0.446 | 0.419 | 0.231 |
| Handwriting | 0.319 | 0.218 | 0.294 | 0.308 | 0.137 | 0.270 | 0.306 | 0.433 | 0.327 | 0.311 | 0.251 | 0.567 | 0.104 | 0.286 |
| Heartbeat | 0.707 | 0.654 | 0.771 | 0.722 | 0.673 | 0.663 | 0.772 | 0.790 | 0.772 | 0.732 | 0.722 | 0.718 | 0.746 | 0.717 |
| Libras | 0.850 | 0.778 | 0.906 | 0.850 | 0.492 | 0.511 | 0.889 | 0.883 | 0.794 | 0.382 | 0.519 | 0.906 | 0.817 | 0.870 |
| LSST | 0.453 | 0.463 | 0.627 | 0.411 | 0.750 | 0.814 | 0.456 | 0.666 | 0.464 | 0.761 | 0.761 | 0.632 | 0.491 | 0.551 |
| MotorImagery | 0.610 | 0.530 | 0.620 | 0.500 | 0.540 | 0.680 | 0.560 | 0.650 | 0.500 | 0.610 | 0.600 | 0.530 | 0.510 | 0.500 |
| NATOPS | 0.767 | 0.767 | 0.844 | 0.828 | 0.756 | 0.810 | 0.917 | 0.906 | 0.917 | 0.833 | 0.756 | 0.885 | 0.794 | 0.883 |
| PEMS-SF | 0.786 | 0.792 | 0.919 | 0.896 | 0.844 | 0.865 | 0.891 | 0.867 | 0.873 | 0.844 | 0.809 | 0.856 | 0.925 | 0.711 |
| PenDigits | 0.936 | 0.957 | 0.948 | 0.972 | 0.894 | 0.973 | 0.973 | 0.983 | 0.974 | 0.984 | 0.974 | 0.996 | 0.855 | 0.977 |
| PhonemeSpectra | 0.166 | 0.180 | 0.261 | 0.233 | 0.119 | 0.087 | 0.131 | 0.176 | 0.113 | 0.146 | 0.081 | 0.284 | 0.155 | 0.151 |
| RacketSports | 0.836 | 0.704 | 0.803 | 0.796 | 0.684 | 0.851 | 0.816 | 0.842 | 0.770 | 0.855 | 0.757 | 0.928 | 0.842 | 0.803 |
| SelfRegulationSCP1 | 0.698 | 0.669 | 0.785 | 0.840 | 0.795 | 0.904 | 0.934 | 0.884 | 0.915 | 0.908 | 0.795 | 0.866 | 0.846 | 0.775 |
| SelfRegulationSCP2 | 0.556 | 0.483 | 0.578 | 0.478 | 0.528 | 0.596 | 0.603 | 0.600 | 0.517 | 0.539 | 0.506 | 0.514 | 0.489 | 0.539 |
| StandWalkJump | 0.530 | 0.333 | 0.600 | 0.400 | 0.400 | 0.787 | 0.467 | 0.467 | 0.333 | 0.533 | 0.467 | 0.456 | 0.467 | 0.200 |
| UWaveGestureLibrary | 0.769 | 0.916 | 0.878 | 0.909 | 0.838 | 0.888 | 0.867 | 0.941 | 0.844 | 0.863 | 0.828 | 0.944 | 0.762 | 0.903 |
| Avg. Accuracy | 0.675 | 0.605 | **0.728** | 0.672 | 0.599 | 0.695 | 0.707 | 0.725 | 0.654 | 0.673 | 0.601 | 0.704 | 0.635 | 0.609 |

# E. Pretraining Dataset

We trained ZEUS on a diverse corpus of real and synthetic time series, comprising approximately 300B observations. We developed the Aegis-Syn synthetic dataset, enhancing KernelSynth dataset (Ansari et al., 2024) with additional diverse patterns. In training, synthetic data comprises roughly 10% of the sampled sequences. To mitigate the imbalance in pattern distributions across the training data, we adopt the balanced sampling strategy (Shao et al., 2025a) rather than assigning fixed probabilities per dataset. This promotes faster convergence and enhances model performance. To prevent data leakage, we exclude all evaluation datasets from pretraining data.

*Table 11.* Synthetic pattern components and their sampling distributions in Aegis-Syn.

| Type | Pattern | Sampling Prob. | Typical Real-world Patterns |
|---|---|---|---|
| Seasonality | Sine | 0.23 | Smooth periodic behavior (e.g., temperature, seasonal demand) |
| | Triangle | 0.13 | Symmetric rise–fall cycles (e.g., tides, workload oscillations) |
| | Sawtooth | 0.10 | Gradual accumulation with abrupt reset (e.g., queues, buffers) |
| | Square | 0.07 | Binary on–off switching (e.g., machine states) |
| | Soft Square | 0.03 | Smoothed regime switching (e.g., controlled systems) |
| | Step | 0.06 | Piecewise-constant shifts (e.g., policy or tariff changes) |
| | Exponential Sawtooth | 0.06 | Accelerating growth before reset (e.g., congestion buildup) |
| | Half-rectified Sine | 0.04 | Positive-only activations (e.g., solar generation) |
| | Full-rectified Sine | 0.03 | Magnitude-only oscillations (e.g., vibration energy) |
| | Amplitude Modulation | 0.05 | Periodic signals with varying intensity (e.g., demand volatility) |
| | Frequency Modulation | 0.05 | Non-stationary periodicity (e.g., drifting rhythms) |
| | Sparse Event (0-1) | 0.08 | Rare discrete events (e.g., faults, alarms) |
| | Pulse | 0.07 | Short-duration impulse signals (e.g., ECG, control inputs) |
| Trend | Linear | 0.50 | Long-term monotonic increase or decrease (e.g., growth, decay) |
| | Logistic | 0.30 | Saturating growth dynamics (e.g., adoption curves) |
| | Long-period Sinusoidal | 0.20 | Slow oscillatory trends (e.g., climate regimes) |
| Noise | Gaussian | 0.60 | Homoscedastic measurement noise (e.g., sensor and acquisition noise) |
| | ARMA | 0.40 | Temporally correlated noise (e.g., system inertia, residual dynamics) |
| Anomaly | None | 0.50 | Normal operating conditions |
| | Point Anomaly | 0.20 | Isolated spikes or drops (e.g., sensor glitches) |
| | Event Anomaly | 0.15 | Short-term level shifts (e.g., incidents, outages) |
| | Contextual Anomaly | 0.10 | Variance or noise regime changes (e.g., instability periods) |
| | Mixed Anomaly | 0.05 | Combination of multiple anomaly types |

### E.1. Real-World Datasets

The real-world data used in our study primarily come from two major sources. The detailed dataset composition is provided in Table 12 and 13.

1. **Chronos Datasets** (Ansari et al., 2024) contain 94B data points. We do not employ the TSMixup data augmentation technique proposed in Chronos.

2. **GiftEvalPretrain Datasets** (Aksu et al., 2024) is the pretraining dataset collection used in the GIFT-Eval benchmark. It comprises 71 univariate and 17 multivariate datasets spanning seven domains and 13 temporal frequencies, totaling 4.5 million time series and 230B data points. This dataset does not overlap with the test sets of the GIFT-Eval benchmark.

### E.2. Aegis-Syn Dataset

We developed Aegis-Syn, a synthetic dataset designed to encompass diverse common patterns in time series data. Influenced by the pioneering KernelSynth (Ansari et al., 2024), the current mainstream approach for synthetic time series generation in model pretraining is based on Gaussian processes (GPs) (Shao et al., 2025a; Wang et al., 2025; Auer et al., 2025). While effective for modeling smooth and stationary dynamics, GPs are inherently ill-suited for representing non-smooth and discontinuous structures that frequently arise in practice, such as abrupt traffic surges or sudden drops in energy consumption.

To overcome this limitation, we construct a component bank that explicitly models diverse temporal structures commonly observed in real-world time series. The bank includes long-term trend components (e.g., linear drift and logistic growth), periodic patterns with both smooth and non-smooth waveforms (e.g., sinusoidal and square-like signals), stochastic noise components capturing random fluctuations, and anomaly components modeling irregular behaviors such as point anomalies and event-level deviations. The complete set of components and their corresponding sampling probabilities are summarized in Table 11. Based on this component bank, each synthetic time series is generated by sampling and composing components according to predefined probability distributions, following the generative procedure described in Algorithm 1.

---

**Algorithm 1** Synthetic Time Series Generation in Aegis-Syn

---

1: **INPUT:** Sequence length $T$, component bank $\mathcal{B}$, sampling distributions $\mathcal{P}$
2: **OUTPUT:** Synthetic time series $\mathbf{x} \in \mathbb{R}^T$
3: **Initialize** $\mathbf{x} \leftarrow \mathbf{0}$
4: // Trend sampling
5: Sample trend type $b_{\text{trend}} \sim \mathcal{P}_{\text{trend}}$ from $\mathcal{B}_{\text{trend}}$
6: Sample trend parameters $\theta_{\text{trend}}$
7: Generate trend component $\mathbf{x}_{\text{trend}} \leftarrow f(b_{\text{trend}}, \theta_{\text{trend}}, T)$
8: $\mathbf{x} \leftarrow \mathbf{x} + \mathbf{x}_{\text{trend}}$
9: // Periodic pattern sampling
10: Sample number of waveforms $K \sim \mathcal{P}_K$
11: **for** $k = 1, \cdots, K$ **do**
12:     Sample waveform type $b_k \sim \mathcal{P}_{\text{wave}}$ from $\mathcal{B}_{\text{wave}}$
13:     Sample waveform parameters $\theta_k = \{\omega_k, \phi_k, a_k\}$
14:     Generate periodic component $\mathbf{x}_{\text{periodic}}^{(k)} \leftarrow f(b_k, \theta_k, T)$
15:     $\mathbf{x} \leftarrow \mathbf{x} + \mathbf{x}_{\text{periodic}}^{(k)}$
16: **end for**
17: // Noise injection
18: Sample noise type $b_{\text{noise}} \sim \mathcal{P}_{\text{noise}}$ from $\mathcal{B}_{\text{noise}}$
19: Sample noise parameters $\theta_{\text{noise}}$
20: Generate noise component $\mathbf{x}_{\text{noise}} \leftarrow g(b_{\text{noise}}, \theta_{\text{noise}}, T)$
21: $\mathbf{x} \leftarrow \mathbf{x} + \mathbf{x}_{\text{noise}}$
22: // Anomaly injection
23: Sample anomaly type $b_{\text{anom}} \sim \mathcal{P}_{\text{anomaly}}$ from $\mathcal{B}_{\text{anomaly}}$
24: **if** $b_{\text{anom}} \neq \text{NONE}$ **then**
25:     Sample anomaly parameters $\theta_{\text{anom}}$
26:     Generate anomaly component $\mathbf{x}_{\text{anom}} \leftarrow h(b_{\text{anom}}, \theta_{\text{anom}}, T)$
27:     $\mathbf{x} \leftarrow \mathbf{x} + \mathbf{x}_{\text{anom}}$
28: **end if**
29: **return** $\mathbf{x}$

---

# F. Limitations and Future Work

**Multivariate Correlations** Like most pre-trained models and TSFMs, ZEUS focuses on univariate time series and handle multivariate time series with channel independent strategy, which has proven effective in practice, as evidenced by the experimental results and prior work such as PatchTST (Nie et al., 2023). Nevertheless, there exist scenarios where incorporating multivariate correlations can lead to significant improvements for downstream tasks. In that cases, adaptation techniques (e.g., CoRA (Qin et al., 2025)) can be adopted. Future work will explore explicitly incorporating multivariate correlations into this framework.

**Generalization to Broader Tasks** At present, ZEUS is primarily evaluated on five categories of downstream tasks that have been extensively explored in previous studies. On the one hand, some important downstream tasks, such as time series segmentation and causal discovery, are not yet well supported by the current framework. On the other hand, several compatible and promising tasks, such as irregular time series forecasting (forecasting tasks with structural missingness), remain unexplored in our evaluation. We leave these directions for future work.

**Challenges in Classification** Time series classification in a fully non-parametric setting remains challenging. While ZEUS with a 1-NN classifier shows promising performance, it is not yet able to outperform state-of-the-art baselines with full-shot training. This can be mainly attributed to two reasons. First, some UEA datasets contain extremely short ($< 30$) and high-dimensional ($> 100$) sequences, whereas ZEUS is better suited for long, univariate time series. Second, the label semantics in classification datasets are highly heterogeneous and strongly task-dependent. To address these issues, we explored contrastive learning (Chen et al., 2020) and SwAV-style prototype contrastive learning (Caron et al., 2020; Li et al., 2021), using joint training or post-training strategies to encourage more clustered representations in ZEUS. However, the

results were not satisfactory. We plan to continue this line of investigation in future work.

## G. Case Study

**Forecasting Showcases**   Figure 10 presents forecast examples from ZEUS. The first eight examples illustrate representative patterns from the Aegis-Syn dataset, while the remaining examples demonstrate the zero-shot forecasting capability of ZEUS. In all cases, ZEUS is able to produce accurate forecasts that closely follow the underlying temporal dynamics.

**Reconstruction Showcases**   Figure 11 presents zero-shot reconstruction examples from ZEUS. For clarity, we primarily present the most challenging cases with a single large contiguous missing block. The first four examples are sampled from imputation benchmarks, demonstrating that the model can accurately reconstruct missing segments and preserve the underlying temporal patterns. The remaining four examples are from the UCR Anomaly Archive, showing that the model is able to precisely localize anomalous regions.

**Clustering Analysis for Classification**   Figure 12 visualizes the representations of ZEUS on the UEA classification datasets after dimensionality reduction using t-SNE (van der Maaten & Hinton, 2008) and PCA. As can be observed, the t-SNE projections exhibit clear clustering patterns, while the PCA projections reveal a coherent low-dimensional manifold structure, indicating that ZEUS learns discriminative and structurally meaningful representations.

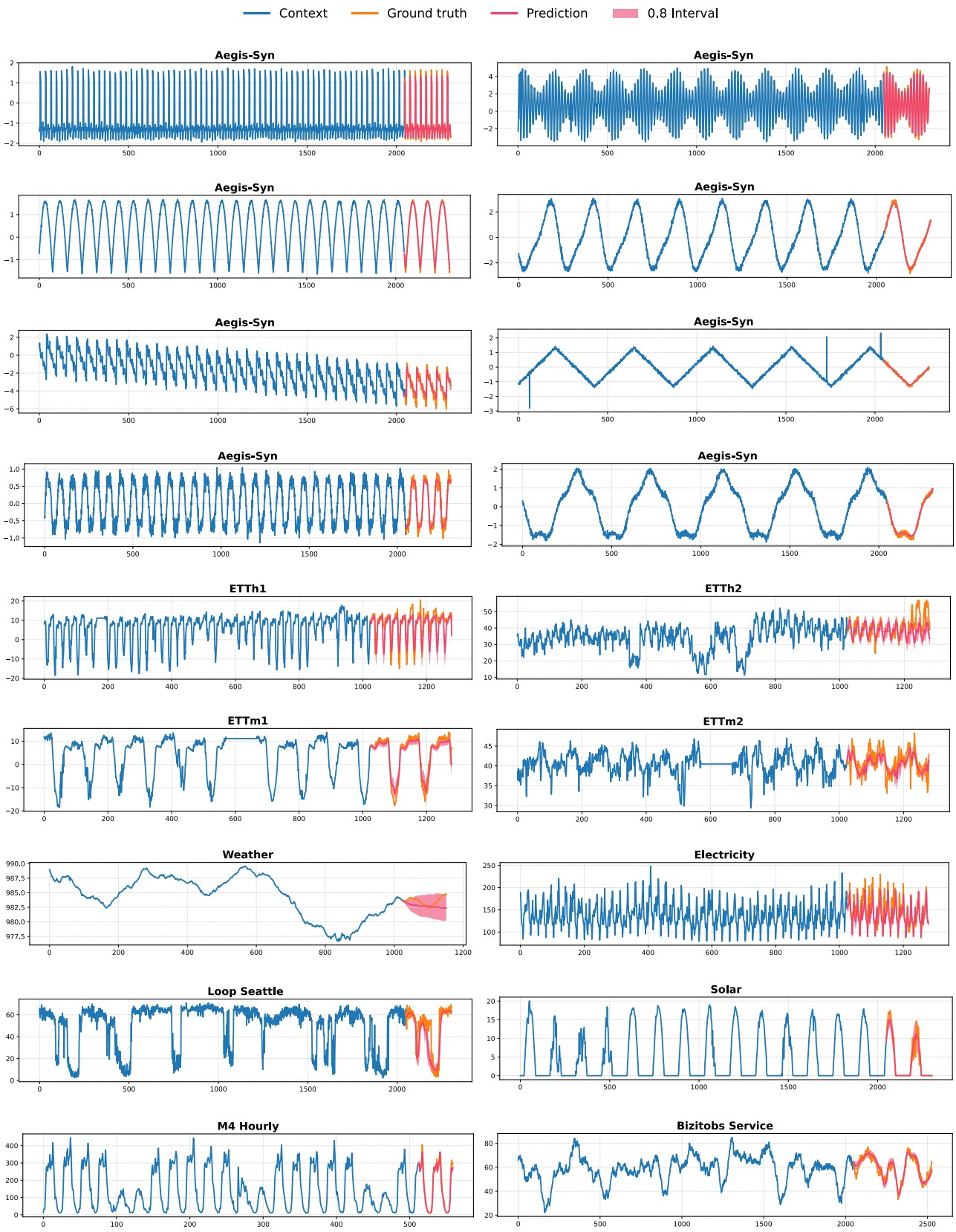

*Figure 10.* Example of forecasts from ZEUS.

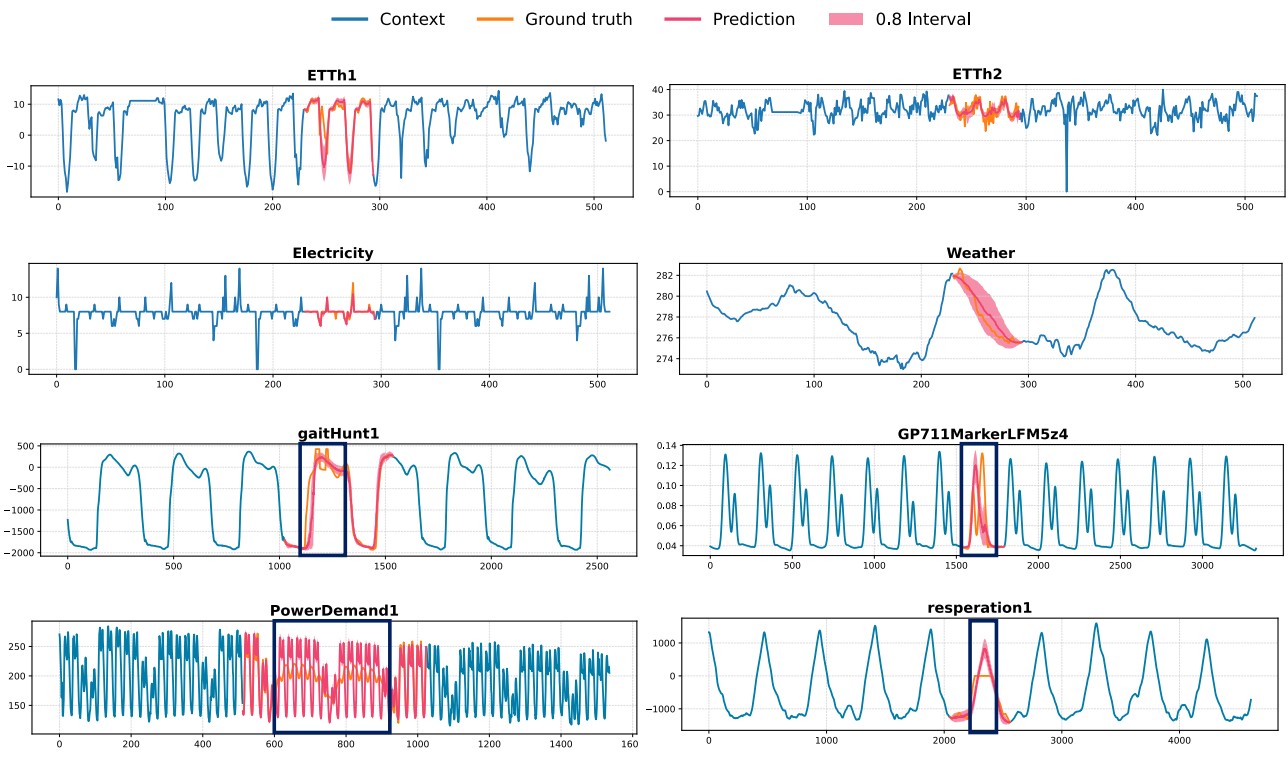

*Figure 11.* Zero-shot examples of reconstruction by ZEUS. *Blue boxes* denote the anomalies identified through reconstruction.

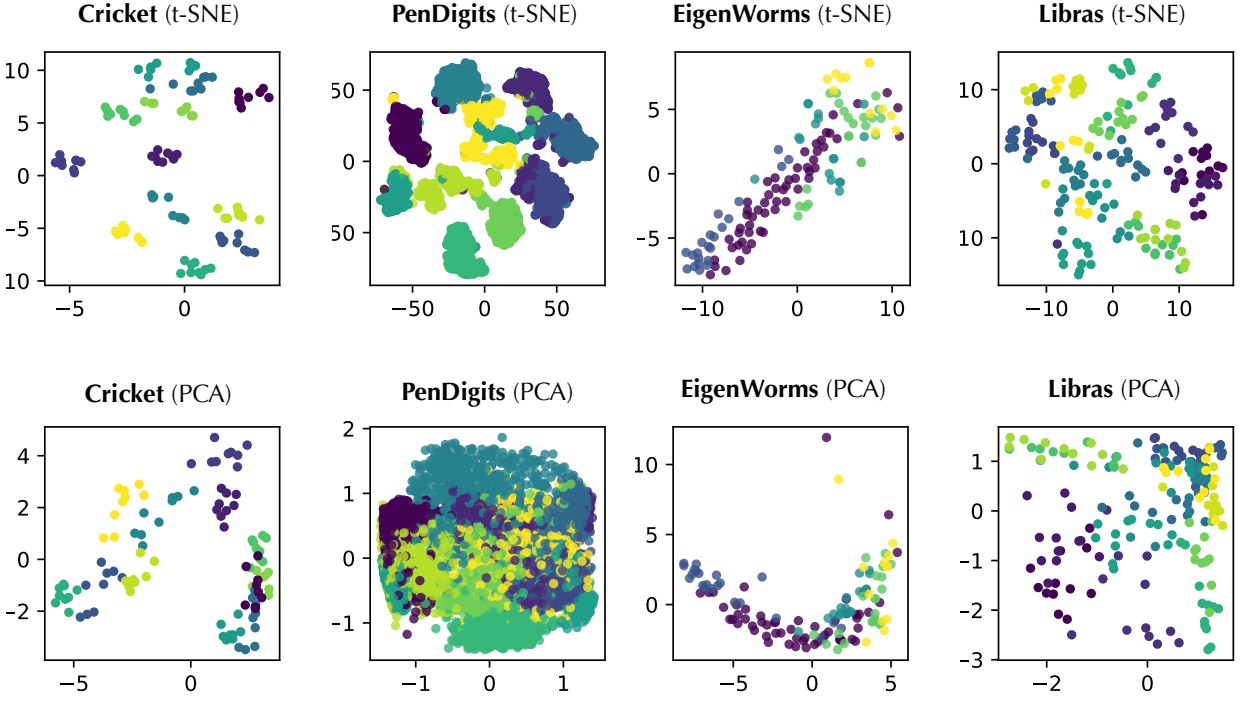

*Figure 12.* Visualization of representations learned by ZEUS on the UEA datasets.

*Table 12.* Statistics of the pretraining datasets.

| Dataset | Domain | Frequency | # Time Series | # Observation |
|---|---|---|---|---|
| Mexico City Bikes | Transport | H | 494 | 38,687,004 |
| Solar | Energy | 5T, H | 5,166 | 1,085,664,720 |
| Spanish Energy and Weather | Energy | H | 1 | 35,064 |
| Taxi | Transport | 30T, H | 4,856 | 3,346,350 |
| USHCN | Climate | D | 6,090 | 235,016,970 |
| Weatherbench | Nature | H, D, W | 225,280 | 79,375,265,520 |
| Wiki (100k) | Web | D | 100,000 | 274,100,000 |
| Wind Farms | Energy | H, D | 100,000 | 856,800,000 |
| KDD Cup 2018 | Energy | H | 270 | 2,897,004 |
| London Smart Meters | Energy | 30T | 5,560 | 166,528,896 |
| M4 | Econ/Fin | D, H, M, W | 52,000 | 50,318,646 |
| Pedestrian Counts | Transport | H | 66 | 3,132,346 |
| Rideshare | Transport | H | 2,304 | 859,392 |
| Temperature-Rain | Nature | D | 32,072 | 22,290,040 |
| Uber TLC | Transport | H, D | 262 | 1,174,932 |
| BDG-2 Panther | Energy | H | 105 | 919,800 |
| BDG-2 Fox | Energy | H | 135 | 2,324,568 |
| BDG-2 Rat | Energy | H | 280 | 4,728,288 |
| BDG-2 Bear | Energy | H | 91 | 1,482,312 |
| Low Carbon London | Energy | H | 713 | 9,543,348 |
| SMART | Energy | H | 5 | 95,709 |
| IDEAL | Energy | H | 219 | 1,265,672 |
| Sceaux | Energy | H | 1 | 34,223 |
| Borealis | Energy | H | 15 | 83,269 |
| Buildings900K | Energy | H | 1,792,328 | 15,702,590,000 |
| CMIP6 | Climate | 6H | 1,351,680 | 1,973,453,000 |
| ERA5 | Climate | H | 245,760 | 2,146,959,000 |
| Azure VM Traces 2017 | CloudOps | 5T | 159,472 | 885,522,908 |
| Borg Cluster Data 2011 | CloudOps | 5T | 143,386 | 537,552,854 |
| Alibaba Cluster Trace 2018 | CloudOps | 5T | 58,409 | 95,192,530 |
| Taxi | Transport | 30T | 67,984 | 54,999,060 |
| Wiki-Rolling | Web | D | 47,675 | 40,619,100 |
| M5 | Sales | D | 30,490 | 58,327,370 |
| LargeST | Transport | 5T | 42,333 | 4,452,510,528 |
| PEMS03 | Transport | 5T | 358 | 9,382,464 |
| PEMS04 | Transport | 5T | 307 | 5,216,544 |
| PEMS07 | Transport | 5T | 883 | 24,921,792 |
| PEMS08 | Transport | 5T | 170 | 3,035,520 |
| PEMS Bay | Transport | 5T | 325 | 16,937,700 |
| Los-Loop | Transport | 5T | 207 | 7,094,304 |
| Beijing Subway | Transport | 30T | 276 | 248,400 |
| SHMetro | Transport | 15T | 288 | 1,934,208 |
| HZMetro | Transport | 15T | 80 | 146,000 |
| Q-Traffic | Transport | 15T | 45,148 | 264,386,688 |
| Subseasonal | Climate | D | 862 | 14,097,148 |
| Subseasonal Precipitation | Climate | D | 862 | 9,760,426 |

*Table 13.* Statistics of the pretraining datasets.

| Dataset | Domain | Frequency | # Time Series | # Observation |
|---|---|---|---|---|
| Covid19 Energy | Energy | H | 1 | 31,912 |
| GEF12 | Energy | H | 20 | 788,280 |
| GEF14 | Energy | H | 1 | 17,520 |
| GEF17 | Energy | H | 8 | 140,352 |
| PDB | Energy | H | 1 | 17,520 |
| BDG-2 Hog | Energy | H | 24 | 421,056 |
| BDG-2 Bull | Energy | H | 41 | 719,304 |
| BDG-2 Cockatoo | Energy | H | 1 | 17,544 |
| ELF | Energy | H | 1 | 21,792 |
| Wind Power | Energy | 4S | 1 | 7,397,147 |
| Solar Power | Energy | 4S | 1 | 7,397,222 |
| Oikolab Weather | Climate | H | 8 | 800,456 |
| Elecdemand | Energy | 30T | 1 | 17,520 |
| Covid Mobility | Transport | D | 362 | 148,602 |
| Kaggle Web Traffic Weekly | Web | W | 145,063 | 16,537,182 |
| Extended Web Traffic | Web | D | 145,063 | 370,926,091 |
| M1 Yearly | Econ/Fin | Y | 106 | 3,136 |
| M1 Quarterly | Econ/Fin | Q | 198 | 9,854 |
| M1 Monthly | Econ/Fin | M | 617 | 44,892 |
| M3 Yearly | Econ/Fin | Y | 645 | 18,319 |
| M3 Quarterly | Econ/Fin | Q | 756 | 37,004 |
| M3 Monthly | Econ/Fin | M | 1,428 | 141,858 |
| M3 Other | Econ/Fin | Q | 174 | 11,933 |
| NN5 Daily | Econ/Fin | D | 111 | 81,585 |
| NN5 Weekly | Econ/Fin | W | 111 | 11,655 |
| Tourism | Econ/Fin | Y, Q, M | 1212 | 150,822 |
| CIF 2016 | Econ/Fin | M | 72 | 6,334 |
| Traffic Weekly | Transport | W | 862 | 82,752 |
| Traffic Hourly | Transport | H | 862 | 14,978,112 |
| Australian Electricity Demand | Energy | 30T | 5 | 1,153,584 |
| Sunspot | Nature | D | 1 | 73,894 |
| Vehicle Trips | Transport | D | 329 | 32,512 |
| Weather | Climate | D | 3,010 | 42,941,700 |
| FRED MD | Econ/Fin | M | 107 | 76,612 |
| Bitcoin | Econ/Fin | D | 18 | 74,824 |
| KDD Cup 2022 | Energy | 10T | 134 | 4,727,519 |
| GoDaddy | Econ/Fin | M | 3,135 | 128,535 |
| Favorita Sales | Sales | D | 111,840 | 139,179,538 |
| Favorita Transactions | Sales | D | 54 | 84,408 |
| China Air Quality | Nature | H | 437 | 5,739,234 |
| Beijing Air Quality | Nature | H | 12 | 420,768 |
| Residential Load Power | Energy | T | 271 | 145,994,559 |
| Residential PV Power | Energy | T | 233 | 125,338,950 |
| CDC Fluview ILINet | Healthcare | W | 75 | 63,903 |
| CDC Fluview WHO NREVSS | Healthcare | W | 74 | 41,760 |
| Project Tycho | Healthcare | W | 1,258 | 1,377,707 |

