# OpenReview forum: "Zeus: Towards Tuning-Free Foundation Model for Time Series Analysis"
_ICML.cc/2026/Conference — ICML 2026 regular_

### Official Review · Reviewer_de2n · 2026-03-07

**Soundness:** 3
**Presentation:** 3
**Significance:** 3
**Originality:** 3
**Overall Recommendation:** 5
**Confidence:** 3

**Summary:**

This paper addresses the core problem that existing Time Series Foundation Models (TSFMs) fail to achieve genuine cross-task generalization. Existing models suffer from two fundamental limitations: patch-level tokenization sacrifices point-level temporal precision, leading to degraded performance on tasks such as imputation and anomaly detection; and a single BERT-style or GPT-style training objective cannot simultaneously cultivate the three heterogeneous capabilities of extrapolation, interpolation, and global abstraction, leaving other tasks dependent on additional fine-tuning. To address these two limitations, this paper proposes ZEUS, which employs a U-shaped multi-scale Transformer to preserve point-level precision while controlling computational overhead, and introduces MOTM, a multi-objective temporal masking strategy that cultivates all necessary capabilities within a single pretraining pass. Experiments demonstrate that ZEUS requires no fine-tuning across five downstream tasks, not only outperforming all other TSFMs comprehensively, but also surpassing fully supervised task-specific models on multiple tasks.

**Compliance With Llm Reviewing Policy:**

Affirmed.

**Final Justification:**

I will keep my score because it is already positive.

**Key Questions For Authors:**

See Strengths and weaknesses above

**Limitations:**

Yes

**Strengths And Weaknesses:**

The point-wise tokenization proposed in ZEUS, combined with a fine→coarse→fine information flow, compensates for the computational burden of overly long point-level token sequences. This design is conceptually sound and well-motivated theoretically. Efficiency analysis shows that ZEUS achieves approximately 3.8× reduction in self-attention FLOPs compared to a vanilla point-tokenized Transformer, which represents a meaningful engineering improvement. I believe this effectively addresses the numerous limitations inherent in patch-wise tokenization adopted by existing TSFMs.
The MOTM strategy proposed in this paper also directly mitigates the inability of a single model architecture to adapt to diverse downstream tasks. This is achieved by mixing predictive, point, multi-block, and single-block masking within a unified pretraining framework, and ablation studies confirm the independent contribution of each masking strategy.
However, one weak point in the experimental section is that, despite the well-motivated design of MOTM, the sampling ratios of each masking strategy (e.g., predictive at 0.3, point at 0.1) are treated as hyperparameters, and the paper provides no systematic sensitivity analysis of these ratios.

---

> ### Author Rebuttal · Authors · 2026-03-30
>
> *We sincerely thank Reviewer de2n for their insightful comments. The following addresses their concerns and provides answers to their questions.*
>
> ---
>
> 1. **Sensitivity analysis**
>
>    Thank you for your suggestion. We clarify that the masking probabilities form a coupled distribution governing the model’s inductive bias, making a full five-probability grid search computationally prohibitive. Instead, we conduct a directional sensitivity analysis with three perturbation scenarios: (1) **Predictive-heavy**: increasing the proportion of predictive masking, (2) **Local-heavy**: increasing point and mult-block masking, and (3) **Structural-heavy**: increasing single-block and mixed masking. We evaluate their impact across three representative tasks, including forecasting (measured by MSE), imputation (MSE), and classification (accuracy).
>
>    The results below show that emphasizing certain masks improves some tasks but degrades others (e.g., local-heavy benefits imputation but harms forecasting). Crucially, our default setting achieves a balance across tasks, indicating robustness and low sensitivity to probability variations.
>
>    |       Type       | Predictive Mask | Point Mask | Single-Block Mask | Multi-Block Mask | Mixed Mask | Forecasting (MSE) | Imputation (MSE) | Classification (Accuracy) |
>    | :--------------: | :-------------: | :--------: | :---------------: | :--------------: | :--------: | :---------------: | :--------------: | :-----------------------: |
>    |   **Original**   |       0.3       |    0.1     |        0.1        |       0.2        |    0.3     |       0.274       |      0.054       |           0.675           |
>    | Predictive-heavy |       0.5       |    0.05    |       0.05        |       0.15       |    0.25    |       0.270       |      0.058       |           0.668           |
>    |   Local-heavy    |       0.2       |    0.5     |       0.05        |       0.3        |    0.2     |       0.279       |      0.052       |           0.673           |
>    | Structural-heavy |       0.2       |    0.05    |       0.25        |       0.1        |    0.4     |       0.276       |      0.058       |           0.678           |
>
> ---
>
> *Thank you for your time and efforts. We hope this has addressed your concerns and answered your questions. Please don’t hesitate to reach out if you have any further questions.*

---

> > ### Author Rebuttal · Reviewer_de2n · 2026-04-04
> >
> > Thanks for the excellent rebuttal. I will keep my score since it already reflect a positive assessment of the work.

---

### Official Review · Reviewer_ErLM · 2026-03-11

**Soundness:** 3
**Presentation:** 4
**Significance:** 3
**Originality:** 3
**Overall Recommendation:** 5
**Confidence:** 5

**Summary:**

This paper presents Zeus, a unified tuning-free time series foundation model. Previous TSFMs require task-specific fine-tuning to perform zero-shot across multiple tasks, and Zeus bridges this gap by addressing two key challenges. First, to reconcile point-level granularity with long-sequence scalability, Zeus incorporates a U-shaped multi-scale architecture, effectively balancing fine-grained modeling and efficiency. Second, to handle varying inductive biases across different tasks, the paper introduces MOTM, a unified strategy that enables different tasks within a single framework. Extensive experiments across five tasks demonstrate the effectiveness of Zeus in tuning-free settings, highlighting its potential as a general-purpose TSFM.

**Compliance With Llm Reviewing Policy:**

Affirmed.

**Final Justification:**

Rebuttal addressed my concerns, and I would remain my suggestion to accept the paper.

**Key Questions For Authors:**

Apart from response to the weakness points, kindly help to address below questions:

1. There are many possible pooling (unpooling) techniques. Why was the specific pooling method used in the paper chosen? What advantages does it offer compared with the alternatives?
2. Have the authors exploresd training models of other sizes? Would increasing the model size further lead to performance improvements (i.e., a scaling law)?
3. In Table 11, how are the proportions of different patterns in the Aegis-Syn dataset determined?

**Limitations:**

yes

**Strengths And Weaknesses:**

S1. The paper is very well written and easy to follow, and it also includes an informative and well-organized appendix.

S2. This work addresses an important problem in time series community—the multi-task compatibility of TSFMs. Although many existing methods claim to be foundation models, they focus only on zero-shot forecasting. Conversely, Zeus is the first TSFM to support multiple downstream tasks without task-specific fine-tuning. Extensive experiments on five benchmarks were conducted to validate the effectiveness of the proposed method, and the performance of Zeus is truly impressive.

S3. The proposed architecture effectively achieves a balance between fine-grained modeling and long-context scalability.

W1 For multi-task end-to-end baselines, representation learning models (e.g., TS2Vec [1] and TSLANet [2]) are not included in the related work section and experiments.

W2 As several existing models such as TTM [3] also adopt multi-scale architectures, clarifying the distinctions between Zeus and these multi-scale approaches would  further strengthen the paper’s contribution.

W3 Regarding MOTM, the details of the mixed masking strategy are not discussed in the paper.

W4 Some minor suggestions:
	1. In Figure 4, the legend colors appear to be inconsistent with those in the figure.
	2. In Section 2.2 (line 93), "increaingly" should be increasingly.
	3. In Table 1, "0..394" should be 0.394
[1] Yue, Z., et al. (2022). Ts2vec: Towards universal representation of time series. In _Proceedings of the AAAI conference on artificial intelligence_ (Vol. 36, No. 8, pp. 8980-8987).
[2] Eldele, E., et al. (2024). TSLANet: Rethinking Transformers for Time Series Representation Learning. In _International Conference on Machine Learning_ (pp. 12409-12428). PMLR.
[3] Ekambaram, V., et al. (2024). Tiny time mixers (ttms): Fast pre-trained models for enhanced zero/few-shot forecasting of multivariate time series. _Advances in Neural Information Processing Systems_, _37_, 74147-74181.

---

> ### Author Rebuttal · Authors · 2026-03-30
>
> *We sincerely thank Reviewer ErLM for their insightful comments. Below we address the key concerns.*
>
> ---
>
> 1. **Representation learning models**
>
>    We agree that representation learning models such as TS2Vec and TSLANet are important. We did not include these models because the representation learning paradigm is better suited to tasks such as classification, and typically shows limited performance or are not directly applicable on forecasting and imputation tasks. This is consistent with prior work (e.g., MOMENT), where such models are evaluated as classification baselines. We therefore report classification results below, where Zeus achieves competitive performance, demonstrating its effectiveness even in scenarios where representation learning methods are typically strong. We also agree that discussing these methods would improve the completeness of the paper, and we will include them in the related work.
>
>    |Metric| Zeus (1-NN) | Zeus (LP) | T-Loss (NeurIPS'19) | TS-TCC (IJCAI'21) | TS2Vec | TSLANet |
>    | :-: | :-: | :-: | :-: | :-: | :-: | :-: |
>    |Avg. Accuracy|0.675| **0.728** |0.642|0.651| 0.680  |  0.721  |
>
> 2. **Distinctions from other multi-scale architectures**
>
>    We thank the reviewer for this insightful suggestion. Although multi-scale designs have been explored by task-specific models (e.g., Pathformer, TimeMixer), most TSFMs remain single-scale. Only a few exceptions explored multi-scale structures, and Zeus differs from these works primarily in **how information flows across scales**. Specifically, TTM follows a **fine-to-coarse** paradigm, progressively aggregating local patches into higher-level representations, whereas Kairos introduces a mixture-of-size patching strategy, partitioning patches in a **coarse-to-fine** manner. In contrast, Zeus introduces a **fine-to-coarse-to-fine multi-scale architecture**, where representations are first aggregated from fine to coarse scales and then refined back to finer resolutions. This design preserves point-level details while simultaneously enhancing long-sequence scalability. We will further clarify this distinction in the revised version.
>
> 3. **Details of mixed masking strategy**
>
>    We provide detailed descriptions of the mixed masking strategy in Appendix A.3. In practice, each mixed masking instance is constructed by combining two masking strategies, where simpler masks (e.g., multi-block and point masking) are paired with more challenging ones (e.g., predictive and single-block masking), ensuring the task is challenging yet learnable.
>
> 4. **Minor suggestions**
>
>    Thank you for pointing out the minor errors and we will revise accordingly.
>
> 5. **Pooling techniques**
>
>    We explored several alternative, including max/average pooling and cross-attention pooling. Empirically, our method achieves the best performance. We attribute this advantage to the  balance between information preservation and flexibility. Specifically, concatenation minimizes information loss by retaining features from all inputs, while the subsequent linear mapping introduces sufficient adaptability to learn effective cross-scale transformations. We will include a brief discussion of these design choices and their empirical impact in the revised version.
>
> 6. **Scaling behavior**
>
>    We agree that understanding scaling behavior is valuable for TSFMs. Due to the multi-scale design, Zeus introduces many hyperparameters (e.g., scale configurations and cross-scale interactions), making exhaustive scaling exploration computationally expensive. Under limited resources, we conducted a preliminary scaling study with different model sizes (～30M, 100M, and 130M parameters). We observe that scaling from 30M to 100M yields consistent improvements, while further increasing to 130M results in marginal gains. This suggests that performance does not scale monotonically with parameter count in our current setting. We hypothesize that this is because performance in multi-scale architectures depends not only on model size but also on the design of scale hierarchies. A systematic study of scaling laws remains an important direction for future work, and we will clarify this in the revised version.
>
> 7. **Details of Aegis-Syn dataset**
>
>    The proportions of different patterns in the Aegis-Syn dataset are determined based on two key principles. (1) Common patterns are assigned higher probabilities, while rare patterns are sampled less frequently. This ensures that the synthetic data distribution better reflects real-world scenarios. (2) Simpler patterns are assigned relatively higher sampling probabilities, whereas more complex or composite patterns are sampled less frequently. This design avoids overloading the model with overly difficult signals in early training, while still exposing it to diverse and challenging cases.
>
> ---
>
> *Thank you for your time and efforts. We hope this has addressed your concerns. Please don’t hesitate to reach out if you have further questions.*

---

> > ### Author Rebuttal · Reviewer_ErLM · 2026-04-02
> >
> > Overall, I consider this paper to be a valuable contribution. As my concerns have been adequately addressed, and after reading the comments from the other reviewers, I support its acceptance and will keep my original score.

---

### Official Review · Reviewer_aL24 · 2026-03-11

**Soundness:** 2
**Presentation:** 3
**Significance:** 2
**Originality:** 2
**Overall Recommendation:** 3
**Confidence:** 4

**Summary:**

This paper introduces ZEUS, a time series foundation model designed to operate in a tuning-free manner across five downstream tasks: forecasting, imputation, anomaly detection, and classification. Experiments on multiple benchmarks demonstrate that ZEUS achieves competitive or state-of-the-art performance without task-specific fine-tuning, outperforming existing foundation models and supervised models on long-term forecasting, probabilistic forecasting, imputation, anomaly detection, and classification tasks.

**Compliance With Llm Reviewing Policy:**

Affirmed.

**Final Justification:**

The proposed tuning-free approach resembles the zero-shot ability of existing foundation models, with the only difference being its capability to perform zero-shot classification. However, the paper lacks designs to support classification and fails to show superior classification results. Further, the paper assembles a few existing methods. Thus, I maintain my score.

**Key Questions For Authors:**

See weaknesses.

**Limitations:**

Yes.

**Strengths And Weaknesses:**

Strengths:
1. The paper validates that point-wise tokenization is more suitable for reconstruction tasks compared to patch-wise tokenization, providing a valuable insight.
2. The multi-scale U-shaped architecture effectively balances fine-grained resolution and long-sequence scalability.
3. The paper achieves state-of-the-art performance.

Weaknesses:
1. Limited novelty: The U-Net architecture is a well-established framework, and the paper simply adopts it without offering additional insights. Similarly, the “Downsampling and Upsampling” approach is a common technique (similar ideas appear in TimeMixer, TimeMixer++), yet the paper does not discuss the differences from such architectures nor presents novel insights. The “Multi-Objective Temporal Masking” essentially integrates normal masking methods of the imputation task without significant innovation.

2. The ablation study only examines the effect of removing individual masking strategies, but it does not explore the sensitivity to hyperparameters.

3. Beyond point-wise and patch-wise tokenization, there is also “variable-wise tokenization” (as in iTransformer). The paper argues that point-wise tokens capture finer-grained information compared to patch-wise tokens, improving reconstruction quality. Would variable-wise tokens be more advantageous for autoregressive tasks, as they may capture more global features suitable for classification? In that context, does point-wise tokenization still hold a clear advantage over variable-wise tokenization?

4. The paper demonstrates strong zero-shot performance across multiple tasks, but it does not analyze how the model behaves when faced with out-of-distribution data or severe distribution shifts. Given that real-world time series often exhibit non-stationarity, it would be helpful to understand whether ZEUS retains its tuning-free advantages under such challenging conditions, or if some lightweight adaptation is still needed.

---

> ### Author Rebuttal · Authors · 2026-03-30
>
> We thank Reviewer aL24 for the insightful comments and address the key concerns below.
>
> 1. **Novelty**
>
>    Our novelty lies not in proposing entirely new primitives, but in **identifying and addressing a critical yet unresolved problem: designing tuning-free time series foundation models (TSFMs)**. We provide a systematic solution motivated by this goal.
>
>    **(1) U-shaped multi-scale hierarchy.** We are the first to introduce a **fine-to-coarse-to-fine multi-scale paradigm** into TSFMs. While the resulting architecture is U-shaped, it is not a direct adoption of U-Net and differs fundementally. In particular, our downsampling strategy is designed as a learnable projection to better preserve temporal patterns across scales, which differs from both U-Net (max pooling/strided convolution) and TimeMixer (average pooling). We further provide a new insight that this fine-to-coarse-to-fine paradigm enables the model to preserve fine-grained details while enhancing scalability, offering a novel solution to the resolution–scalability dilemma.
>
>    **(2) Multi-Objective Temporal Masking (MOTM).** Unlike prior TSFMs that typically use a single objective (e.g., BERT/GPT-style), **we are the first to systematically construct multiple masking schemes for genuine multi-task capability**. Rather than adopting off-the-shelf masking, MOTM is tailored to TSFM pretraining, enabling effective multi-task generalization. For example, our single-block masking follows a geometric distribution, which better reflects real-world missing patterns than the Poisson-based span masking in NLP.
>
>    In summary, **our contribution is not a simple combination, but a fundamental reframing of architecture and training objectives to enable zero-shot multi-task generalization.**
>
> 2. **Sensitivity analysis**
>
>    Since five masking probabilities form a coupled distribution, full grid search is computationally prohibitive. Instead, we conduct a directional sensitivity analysis across three scenarios: (1) **Predictive-heavy**: increasing predictive masking, (2) **Local-heavy**: increasing point and multi-block masking, and (3) **Structural-heavy**: increasing single-block and mixed masking. We evaluate their impact on three representative tasks, including forecasting (measured by MSE), imputation (MSE), and classification (Acc.).
>
>    Results show that emphasizing certain masks benefits some tasks but harms others. Our default setting achieves a balance across tasks, indicating robustness and low sensitivity to variations.
>
>    |Type|Predictive Mask|Point Mask|Single-Block Mask|Multi-Block Mask|Mixed Mask|Forecasting (MSE)|Imputation (MSE)| Classification (Acc.) |
>    |:-:|:-:|:-:|:-:|:-:|:-:|:-:|:-:|:-:|
>    |**Original**|0.3|0.1|0.1|0.2|0.3|0.274|0.054|0.675|
>    |Predictive-heavy |0.5|0.05|0.05|0.15|0.25|0.270|0.058|0.668|
>    |Local-heavy|0.2|0.5|0.05|0.3|0.2|0.279|0.052|0.673|
>    |Structural-heavy |0.2|0.05|0.25|0.1|0.4|0.276|0.058|0.678|
> 3. **Variable-wise tokenization**
>
>    While variable-wise tokenization aggregates global information and may benefit tasks like classification, it requires **fixed input length** (mapping from input length $L$ to fixed dimension $D$), conflicting with TSFMs’ need for **variable-length inputs and flexible horizons**. While resampling could enforce fixed length, it may distort temporal dynamics and degrade model performance. This limitation is also reflected in existing TSFMs, which generally do not adopt variable-wise tokenization. We will discuss the tokenization of TSFM in the revised version.
>
> 4. **Out-of-distribution (OOD) and distribution shift scenarios**
>
>    OOD generalization is precisely the scenario that Zeus is designed for. All experiments in our paper are conducted under zero-shot settings, where Zeus is directly evaluated on unseen datasets, i.e., evaluation benchmarks are inherently OOD. Through large-scale pretraining on diverse data, Zeus learns a wide range of temporal patterns and can generalize to unseen distributions, as evidenced by the SOTA performance across multiple benchmarks.
>
>    We further distinguish two types of distribution shifts:
>
>    **(1) Inter-sample/dataset distribution shift**, which is actually an OOD scenario, as it involves transferring across unseen distributions. Thus, it is naturally handled by Zeus’s zero-shot generalizabiliy. This is evidenced by strong performance on challenging real-world datasets with non-stationary dynamics (e.g., *Saugeen*, *SZTaxi* from GiftEval, and *Weallwalk* from UCR-AD), demonstrating its robustness under practical distribution shifts.
>
>    **(2) Intra-sample distribution shift**, where the future dynamics change due to unobserved exogenous factors, making it inherently unpredictable. This limitation is not specific to Zeus but reflects a fundamental challenge in time series modeling.
>
>    Overall, Zeus can effectively handle OOD and inter-sample shifts by large-scale pretraining, while intra-sample shifts require external information or adaptation.

---

> > ### Author Rebuttal · Reviewer_aL24 · 2026-04-02
> >
> > I would like to thank the authors for the efforts put on the design, experiments, and the rebuttal. After careful consideration, I still decided to maintain my original score due to the fundamental issues lying in the paper. I will leave the final decision to the AC.
> >
> > 1. Despite claiming the novelty lying in the "tuning-free" capability across tasks, the experiments rely on extensive hyperparameter search (e.g., window size). This contradicts to the claim of "tuning-free" stated in the novelty of the paper, and does not sound like "tuning-free". Furthermore, the proposed tuning-free approach underperforms in classification tasks (classification actually does need real "tuning-free", but the proposed approach fails to perform well), and the proposed method lacks targeted designs to improve this capability.
> >
> > 2. The multi-scale architecture also lacks sufficient novelty to be published at a top conference. Specifically, it appears to be an adaptation of existing progressive multi-scale combinded with the U-Net designs to time-series foundation models. It doesn't matter whether this is first aplied to TSFM, as TSFM is only a setting. But, the multi-scale paradigm has been studied before, such as Scaleformer (ICLR 2023).
> >
> > Based on the two key issues related with the claim and the novelty of the paper, I decided to keep my score.

---

> > > ### Author Response · Authors · 2026-04-03
> > >
> > > We sincerely thank the reviewer for the response. We respect the reviewer’s perspective and would like to provide further clarification regarding our tuning-free setting and key contribution.
> > >
> > > 1. **Tuning-free setting**
> > >
> > >    We apologize for the potentially misleading use of the term *tuning-free* in our paper. We would like to clarify that in our work, *tuning-free* refers to **not requiring any parameter tuning or retraining of the model backbone across tasks**. In time series analysis tasks, it is generally unavoidable to specify certain inference-time configurations, such as the context length (window size), since they are inherently tied to the formulation of the task. However, these choices do not involve modifying or optimizing the model parameters and no additional training is required.
> > >
> > >    Regarding the concern about hyperparameter search, we would like to clarify that the search is **restricted to inference-time configurations** (e.g., context length), while the backbone remains completely fixed. This practice is consistent with standard experimental protocols in time series analysis [1, 2, 3]. Importantly, such choices do not require expensive retraining or per-task adaptations.
> > >
> > >    Regarding the classification performance, we acknowledge that our method does not outperforms all task-specific baselines, as also discussed in our limitations. However, we would like to emphasize that Zeus under 1-NN setting outperforms all existing multi-task TSFMs, including those with fine-tuning. We believe this is a meaningful result, as it demonstrates that our model learns generalizable representations without any task-specific optimization. We agree that incorporating more targeted designs could further improve classification performance, and this is an important direction for future work (*line 1352-1354*).
> > >
> > >    In summary, our *tuning-free* claim pertains to **eliminating per-task adaptation**, rather than removing all forms of inference configuration. While some task-dependent hyperparameters (e.g., context length) are necessary, they do not undermine the core advantage of avoiding per-task adaptation, which is the central goal of our approach. We will revise the paper to clarifying it as *no-parameter-tuning* where appropriate and avoid confusion.
> > >
> > > 2. **Key contribution**
> > >
> > >    We respect the reviewer’s perspective that our approach builds upon multi-scale ideas rather than introducing entirely new architectural primitives. However, we would like to emphasize that our main contribution lies in **identifying a previously underexplored problem, tuning-free TSFM, and providing a concrete and effective solution to it**.
> > >
> > >    While our approach may not be perfect, it represents a **step toward a more general-purpose and practical TSFM paradigm**, which we believe an important direction for this field. In contrast to prior works that primarily focus on zero-shot forecasting and rely on task-specific adaptations for other downstream tasks, our model demonstrates the ability to handle multiple tasks under a unified, tuning-free framework.
> > >
> > > We sincerely thank the reviewer for the thorough engagement and constructive feedback. We hope that our responses clarify the key concerns and better convey the contribution of our work.
> > >
> > > ---
> > >
> > > **References**
> > >
> > > [1] Shao, Zezhi, et al. "Exploring progress in multivariate time series forecasting: Comprehensive benchmarking and heterogeneity analysis." *IEEE Transactions on Knowledge and Data Engineering* 37.1: 291-305. 2024.
> > >
> > > [2] Li, Zhe, et al. "Tsfm-bench: A comprehensive and unified benchmark of foundation models for time series forecasting." *Proceedings of the 31st ACM SIGKDD Conference on Knowledge Discovery and Data Mining V. 2*. 2025.
> > >
> > > [3] Shi, Xiaoming, et al. "Time-MoE: Billion-Scale Time Series Foundation Models with Mixture of Experts." *The Thirteenth International Conference on Learning Representations*. 2025.

---

### Official Review · Reviewer_e5fm · 2026-03-13

**Soundness:** 3
**Presentation:** 4
**Significance:** 3
**Originality:** 3
**Overall Recommendation:** 4
**Confidence:** 1

**Summary:**

Time Series Foundation Models (TSFMs) aim to create general-purpose models that work across many time-series tasks (forecasting, imputation, anomaly detection, classification). Existing models usually work zero-shot for forecasting but require task-specific fine-tuning for other tasks. Overall, this research's main result is: The paper proposes ZEUS, a tuning-free time series foundation model that performs competitively across multiple tasks without task-specific fine-tuning.

**Compliance With Llm Reviewing Policy:**

Affirmed.

**Final Justification:**

Overall, this research's main result is the proposal of ZEUS, a tuning-free time series foundation model that generalizes across multiple tasks without task-specific fine-tuning. A broad context studied by the study is unified TSFMs for forecasting, imputation, anomaly detection, and classification.

The paper is **technically sound and well-motivated**, with reasonable architectural and training design choices and strong multi-task evaluation. It is also clearly written.

My main concern is **limited novelty**, as the approach largely combines existing ideas. The rebuttal addressed my concerns well and strengthened my confidence in the work’s soundness, though it did not significantly change my view on originality.

Overall, I consider this a **borderline conference paper**, but also a strong **position/system paper** with practical relevance.

**Key Questions For Authors:**

Since most of ZEUS’s training data comes from two dataset collections, could the authors provide more analysis or visualizations of the training data distribution (e.g., domains, frequencies, pattern types) to better illustrate its diversity and how it differs from the evaluation benchmarks?

**Limitations:**

yes

**Strengths And Weaknesses:**

Strengths

1. The paper clearly identifies architectural and training limitations in existing TSFMs and frames the problem of tuning-free multi-task time-series modeling well.

2. The multi-scale U-shaped Transformer is a reasonable design for balancing local temporal fidelity and long-context efficiency.

3. Multi-task pretraining objective, the MOTM strategy is intuitive and aligns with different task requirements (forecasting vs interpolation vs global representation).

4. Broad empirical evaluatio nExperiments cover five downstream tasks and several benchmarks, which strengthens the claim of general-purpose applicability.

5. The paper includes both theoretical and empirical efficiency analysis of the multi-scale architecture.

Weaknesses

1. Limited methodological novelty

Most components (hierarchical Transformers, masked reconstruction, span masking) exist in prior literature. The novelty mainly comes from combining these ideas.

2. Results are reported as single numbers. The paper does not report: variance across runs, stdev,confidence intervals

---

> ### Author Rebuttal · Authors · 2026-03-31
>
> *We sincerely thank Reviewer e5fm for their insightful comments. The following addresses their concerns and provides answers to their questions.*
>
> ---
>
> 1. **Methodological novelty**
>
>    Thank you for raising this important concern. We would like to clarify that our contribution is not a straightforward combination of existing components, but a problem-driven redesign centered on a critical yet unresolved question: **how to build tuning-free time series foundation models (TSFMs)**. This objective fundamentally shapes both our architecture and training strategy.
>
>    **(1) Multi-scale architecture.** Although multi-scale designs have been explored by task-specific models, most TSFMs remain single-scale. Only a few exceptions explored multi-scale structures, and Zeus differs from these works fundamentally in both **information flow and architectural design**. For example, TTM [1] follows a **fine-to-coarse** paradigm, progressively aggregating local patterns, while Kairos [2] adopts a **coarse-to-fine** mixture-of-size patching strategy. In contrast, Zeus is the first to introduce a **fine-to-coarse-to-fine multi-scale architecture**, whch preserves point-level details while simultaneously enhancing long-sequence scalability. Rather than reusing existing multi-scale Transformers, our design is specifically tailored to address the granularity–scalability dilemma in TSFMs.
>
>    **(2) Multi-Objective Temporal Masking (MOTM).** Existing methods typically rely on a single objective (e.g., BERT/GPT-style masked reconstruction). In contrast, we **systematically design and integrate multiple masking schemes to enable genuine multi-task capability**. Rather than adopting off-the-shelf masking, MOTM is tailored to TSFM pretraining, enabling effective multi-task generalization. For example, our single-block masking follows a geometric distribution, better matching real-world missing patterns and more suitable for TSFM pretraining than the Poisson-based span masking used in NLP. Moreover, the combination of masking strategies is carefully designed to balance the inductive biases across different tasks.
>
>    In summary, **our contribution is not a simple combination of existing techniques, but a fundamental reframing of architecture and training objectives to enable zero-shot multi-task generalization.**
>
>
> 2. **Results are reported as single numbers**
>
>    We would like to clarify that, as a foundation model, Zeus's inference process is deterministic. Given fixed model parameters and inputs, repeated evaluations yield identical results. Therefore, reporting variance across runs, standard deviation, or confidence intervals from repeated inference is not applicable in Zeus's evaluation.
>
>    We acknowledge that, for task-specific models, it is common practice to report statistics over multiple training runs to account for randomness in initialization and optimization. However, for large-scale foundation models, retraining multiple times is typically prohibitively expensive in terms of computational resources and time. As a result, it is standard practice in the literature on foundation models to report single-run performance.
>
> 3. **Pretraining data statistics**
>
>    Thank you for this insightful suggestion. We agree that understanding the distribution of the pretraining data is important for interpreting the model’s generalizability. Tables 12 and 13 in the Appendix already provide detailed statistics of the training data. We further analyze the data distribution by domain and sampling frequency, as summarized in the tables below. It can be observed that the raw data distribution is highly imbalanced, with climate-related datasets accounting for the majority of observations, most of which are at hourly frequencies (H/6H). To mitigate the risk of the model being dominated by such data, we adopt a balanced sampling strategy (*Line 1240–1242*). In the revised version, we will include additional visualizations to better illustrate these distributions.
>
>    **Domain**
>
>    |   Domain   | Climate/Nature | Energy | CloudOps | Transport | Others |
>    | :-: | :-: | :----: | :-: | :-: | :-: |
>    | Proportion |     87.5%      |  7.0%  |   3.1%   |   2.1%    |  0.3%  |
>
>    **Frequency**
>
>
>    | Frequency  | 4S/T | H/6H  |  D   | Others (W/M/Q/Y) |
>    | :-: | :--: | :---: | :--: | :-: |
>    | Proportion | 2.2% | 96.7% | 1.1% |       <0.1%       |
>
>    *T includes T, 5T, 10T, 15T, 30T*
>
> ---
>
> *Thank you for your time and efforts. We hope this has addressed your concerns and answered your questions. Please don’t hesitate to reach out if you have any further questions.*
>
> ---
> **Reference:**
>
> [1] Ekambaram, Vijay, et al. "Tiny time mixers (ttms): Fast pre-trained models for enhanced zero/few-shot forecasting of multivariate time series." *Advances in Neural Information Processing Systems* 37 (2024): 74147-74181.
>
> [2] Feng, Kun, et al. "Kairos: Towards Adaptive and Generalizable Time Series Foundation Models." *arXiv preprint arXiv:2509.25826* (2025).

---

> > ### Author Rebuttal · Reviewer_e5fm · 2026-04-02
> >
> > Great! Most of my concerns have been addressed.
> >
> > I also understand the authors’ point regarding evaluation cost and deterministic inference. I have one remaining question: Could the authors provide an analysis of how inference hyperparameters (e.g., context length selection) are chosen, along with an ablation or sensitivity analysis on their impact on inference results? If this is too expensive, the final version should clearly specify all inference-time hyperparameters to ensure reproducibility.
> >
> > I will maintain my original score, as the rebuttal does not materially change my overall assessment of the paper.

---

> > > ### Author Response · Authors · 2026-04-05
> > >
> > > Thank you for the response! We are glad that our previous clarification helped address your concerns. Below, we further elaborate on the new questions regarding inference hyperparameters.
> > >
> > > We have provided detailed inference hyperparameter ranges in Appendix C. For instance, the context length in point forecasting task is selected via hyperparameter search from the set {512,720,1024,2048,3072} (*line 867-868*). Empirically, increasing the context length generally improves model performance. However, the extent of improvement depends strongly on the inherent properties of the data, particularly its stationarity. For example, on the ETTh1 dataset, increasing the context length from 720 to 3072 only brings marginal improvement, whereas the gain is much more significant on the Weather dataset (indicating its underlying patterns are more stationary). In contrast, on ETTh2, extending the context length to 3072 even leads to a performance degradation.
> > >
> > > | Datasets | Metrics | 512   | 720   | 1024  | 2048      | 3072      |
> > > | -------- | ------- | ----- | ----- | ----- | --------- | --------- |
> > > | ETTh1    | MSE     | 0.354 | 0.344 | 0.346 | 0.343     | **0.341** |
> > > |          | MAE     | 0.378 | 0.374 | 0.375 | 0.373     | **0.372** |
> > > | ETTh2    | MSE     | 0.280 | 0.283 | 0.275 | **0.266** | 0.268     |
> > > |          | MAE     | 0.326 | 0.326 | 0.321 | **0.317** | 0.318     |
> > > | Weather  | MSE     | 0.153 | 0.152 | 0.150 | 0.147     | **0.145** |
> > > |          | MAE     | 0.195 | 0.193 | 0.191 | 0.187     | **0.185** |
> > >
> > > Overall, the model exhibits varying sensitivity to hyperparameters across different datasets, which is largely driven by dataset-specific characteristics rather than the model itself. Consequently, analyzing inference hyperparameter sensitivity on a per-dataset basis provides limited insight into the model's robustness. For this reason, we report the general hyperparameter ranges instead of conducting dataset-specific hyperparameter analyses. However, we will include a more detailed discussion of this issue in the revised version.
> > >
> > > We sincerely thank you again for your thoughtful engagement and constructive feedback!

---

### Decision · Program_Chairs · 2026-04-30

**Decision:**

Accept (regular)

**Comment:**

Reviewer e5fm
Overall, this research's main result is the proposal of ZEUS, a tuning-free time series foundation model that generalizes across multiple tasks without task-specific fine-tuning. A broad context studied by the study is unified TSFMs for forecasting, imputation, anomaly detection, and classification.
The paper is technically sound and well-motivated, with reasonable architectural and training design choices and strong multi-task evaluation. It is also clearly written.
While reviewer e5fm main concern was limited novelty, the rebuttal addressed the concerns well and strengthened the reviewer's confidence in the work’s soundness, though it did not significantly change the reviewer's view on originality.

Reviewer aL24
The proposed tuning-free approach resembles the zero-shot ability of existing foundation models, with the only difference being its capability to perform zero-shot classification. However, the paper lacks designs to support classification and fails to show superior classification results. Further, the paper assembles a few existing methods. Thus, I maintain my score.

Reviewer ErLM
Rebuttal addressed my concerns, and I would remain my suggestion to accept the paper.


Reviewer de2n (5)
I will keep my score because it is already positive.

Authors have responded carefully to reviewer aL24 comments.
1. To the reviewer aL24 comment that the method lacks novelty and the method is not tuning free as claimed by the authors, authors clarify that the tuning-free refers to not requiring any parameter tuning or retraining of the model backbone across tasks. It must be noted that certain inference time configurations (context length etc) are inevitable and do not require additional training. Hence, the method can still be considered tuning free, although the inference time configurations can be considered as hyperparameters.
2. To the reviews aL24 comment that the multi-scale architecture also lacks sufficient novelty and it appears to be an adaptation of existing progressive multi-scale combined with the U-Net designs to time-series foundation models, the authors clarify that their main contribution lies in identifying a previously underexplored problem, tuning-free TSFM, and providing a concrete and effective solution to it; and offers step toward a more general-purpose and practical TSFM paradigm - which is a hard problem as time series data lacks a clear dictionary.